# Evolution and compression in LLMs: On the emergence of human-aligned categorization

**Nathaniel Imel**
New York University
n.imel@nyu.edu

**Noga Zaslavsky**
New York University
nogaz@nyu.edu

## Abstract

Converging evidence suggests that human systems of semantic categories achieve near-optimal compression via the Information Bottleneck (IB) complexity-accuracy tradeoff. Large language models (LLMs) are not trained for this objective, which raises the question: are LLMs capable of evolving efficient human-aligned semantic systems? To address this question, we focus on color categorization—a key testbed of cognitive theories of categorization with uniquely rich human data—and replicate with LLMs two influential human studies. First, we conduct an English color-naming study, showing that LLMs vary widely in their complexity and English-alignment, with larger instruction-tuned models achieving better alignment and IB-efficiency. Second, to test whether these LLMs simply mimic patterns in their training data or actually exhibit a human-like inductive bias toward IB-efficiency, we simulate cultural evolution of pseudo color-naming systems in LLMs via a method we refer to as Iterated in-Context Language Learning (IICLL). We find that akin to humans, LLMs iteratively restructure initially random systems towards greater IB-efficiency. However, only a model with strongest in-context capabilities (Gemini 2.0) is able to recapitulate the wide range of near-optimal IB-tradeoffs observed in humans, while other state-of-the-art models converge to low-complexity solutions. These findings demonstrate how human-aligned semantic categories can emerge in LLMs via the same fundamental principle that underlies semantic efficiency in humans.

## 1 Introduction

As large language models (LLMs) become increasingly popular in everyday use, it is crucial to understand how their learning biases and representational capacities align with our own. Here, we investigate this by focusing on a key aspect of human intelligence: the ability to organize information into semantic categories (Croft, 2002; Rosch, 2002; Koch, 2008; Boster, 2005; Majid, 2015; Malt & Majid, 2013). This phenomenon presents two major challenges for AI. First, systems of semantic categories (semantic systems, for short) exhibit both universal patterns and cross-language differences (Berlin & Kay, 1969; Berlin, 1992; Croft, 2002), which LLMs must navigate. Second, LLMs are not grounded in the rich physical and social environment that humans are (Rosch, 1975; Labov, 1973), and it is unclear how these differences affect their ability to learn human-aligned semantic categories. Therefore, in order to understand whether LLMs can efficiently communicate with people and adapt to changing environments and communicative needs in a human-like manner, it is crucial to study whether LLMs are capable of structuring meaning according to the same principles that guide humans.

To address this open gap in our understanding of LLMs, we propose a novel theoretical and cognitively-motivated framework for studying semantic systems in LLMs. We build on the framework of Zaslavsky et al. (2018), which argues that languages efficiently compress meanings into words by optimizing the Information Bottleneck (IB) principle (Tishby et al., 1999), instantiated as a tradeoff between the informational complexity and communicative accuracy of the lexicon. This framework has broad empirical support across human languages (Zaslavsky et al., 2018; 2019; 2021; Mollica et al., 2021; Zaslavsky et al., 2022). Furthermore, Imel et al. (2025) recently showed that a drive for IB-efficiency may be present in the individual inductive biases of human learners. While LLMs are trained on vast amounts of human language data encoded as text, they are not trained with

respect to the IB objective. This raises the question: are LLMs capable of developing IB-efficient human-aligned semantic systems?

We address this open question with an in-depth analysis in the domain of color—a key test case for categorization theories in cognitive science with rarely available human data as well as practical implication for human-LLM interactions (see Section 2.1)—and replicate with LLMs two influential human behavioral experiments (Figure 1). First, we conduct an English color naming experiment (analogous to the human experiment of Lindsey & Brown, 2014), designed to assess the efficiency and human-alignment of the color naming systems of LLMs. Second, we conduct an iterated learning experiment of pseudo color-naming systems (analogous to the human experiment of Xu et al., 2013), designed to reveal implicit inductive learning biases of LLMs by simulating a process of cultural transmission (see Section 2.3). For the latter, we extend Zhu & Griffiths (2024)'s iterated in-context learning (I-ICL) paradigm to iterated in-context *language* learning (IICLL).

Our key findings and contributions are summarized as follows: First, we show that many prominent LLMs struggle to capture the English color naming system, exhibiting a wide range of complexities that are often lower than the complexity of English. However, with increased size and instruction-tuning, LLMs can achieve high English-alignment and IB efficiency. Second, using our IICLL paradigm, we show that LLMs that perform well in the naming task are not merely mimicking patterns in their training data but are actually guided by a human-like inductive bias toward IB-efficiency. Specifically, we show that LLMs iteratively restructure initially random artificial systems towards greater IB-efficiency and increased human-alignment. However, among the models we tested, only the model with strongest in-context capabilities (Gemini 2.0) is able to recapitulate the wide range of near-optimal IB-tradeoffs observed in humans, while other models converge to low-complexity solutions. Finally, we show that Gemini can also develop structured category systems via IICLL in a domain that is qualitatively different from color, suggesting that our result could potentially apply also in other domains.

Taken together, our findings demonstrate how human-aligned semantic categories can emerge in LLMs via the same fundamental principle that underlies semantic efficiency in humans. Importantly, neither humans nor LLMs are explicitly trained for optimizing the IB objective, suggesting that IB-efficiency, and optimal compression more generally, may emerge to support intelligent behavior.

## 2 BACKGROUND AND MOTIVATION

### 2.1 TALKING ABOUT COLOR

For decades, cognitive scientists have used color as an essential tool to study perception and categorization. This is in part due to the unprecedented amount of human behavioral data available for color, which includes research on perceptual space, cross-linguistic semantic variation, category learning, and cultural evolution. Especially relevant to our study is the World Color Survey (WCS) dataset (Cook et al., 2005), which contains color-naming data from 110 non-industrialized languages. Another relevant study by Xu et al. (2013) demonstrated that the cultural transmission of initially random, artificial color-term systems in humans leads to more regular systems that resemble those documented in the WCS dataset. To our knowledge, these two data resources— on actual cross-linguistic naming patterns and on the cultural evolution of category systems—are unique to the domain of color, making it an ideal domain for evaluating how well LLMs align with human behavior. Furthermore, studying color naming in LLMs also has practical implications. Generative AI models, used for tasks like image generation and online product searches, require grounded representations of color language. In order for such models to interact in ways we expect them to, it is crucial to determine the extent to which state-of-the-art LLMs have learned the meaning of color terms that actually align with human naming patterns.

While previous work has shown that human-aligned color representations can be recovered from LLMs (e.g., Abdou et al., 2021; Patel & Pavlick, 2022), there is limited research on model behavior in the context of real-world settings (namely, prompt-based interactions). Marjieh et al. (2024) showed that several recent instruction-tuned models (specifically, GPT-3, its "ChatGPT" variants, GPT-3.5 and GPT-4 (Brown et al., 2020; OpenAI et al., 2024) and Mistral 7B Instruct (Jiang et al., 2023)) can recover the English and Russian color naming systems by prompting the models to label hex codes. In contrast, we test a large set of 39 models with varying sizes and training stages, we

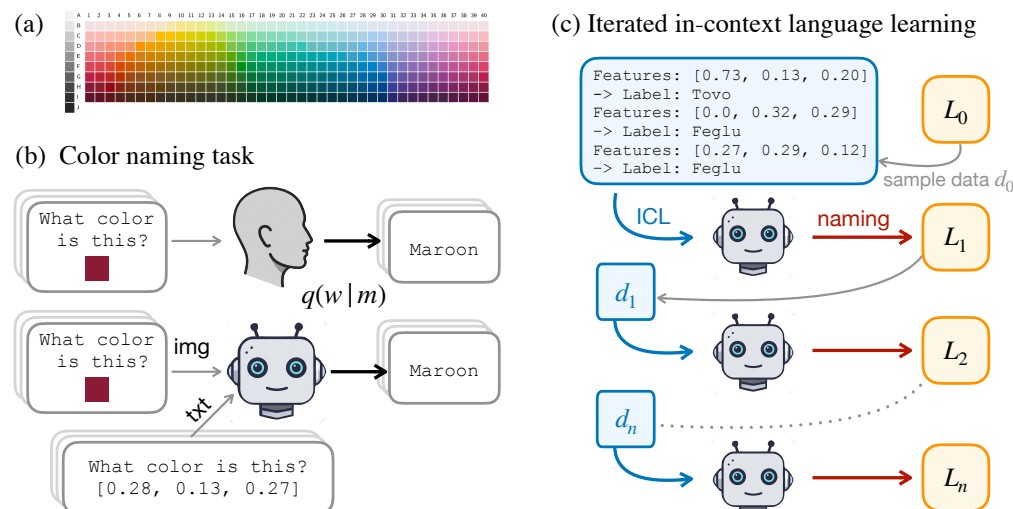

Figure 1: (**a**) The standard WCS color naming grid (Kay et al., 2009). (**b**) Color naming task with humans and LLMs. Multi-modal LLMs can observe colors either via text or images. (**c**) Illustration of the IICLL paradigm. At each generation $t$, an LLM is prompted with a small dataset for ICL, $d_{t-1}$, consisting of pairs of colors and pseudo labels sampled from the previous generation's language, $L_{t-1}$. With these data in context, the LLM performs the naming task for the full space (a).

consider both textual inputs and image inputs (for multi-modal models), we analyze the LLMs' color naming systems through the lens of the IB framework, and we further assess their underlying inductive learning biases in a cultural evolution process via iterated learning. Next, we provide relevant background on the IB and iterated learning frameworks below.

## 2.2 THE INFORMATION BOTTLENECK PRINCIPLE AND SEMANTIC SYSTEMS

The IB framework for semantic systems (Zaslavsky et al., 2018), which we employ in this work, is based on the following communication model: a speaker wishes to communicate a mental representation, or belief state, $m \in \mathcal{M}$, defined as a probability distribution over world states $u \in \mathcal{U}$, by mapping it to a word $w \in \mathcal{W}$ via a stochastic encoder $q(w|m)$. A listener then receives $w$ and interprets the speaker's intended meaning by constructing an estimator $\hat{m}_w$. In the case of color, for example, the world states $\mathcal{U}$ are given by a set of target colors (Figure 1a). To account for perceptual noise, it is assumed that each $m$ is a Gaussian distribution over the perceptual CIELAB color space centered around a corresponding target color (Zaslavsky et al., 2018). That is, each color is mentally represented by the speaker as a Gaussian distribution and the listener's goal is to reconstruct the speaker's belief state over colors.

According to this framework, in order to communicate efficiently, the speaker and listener must jointly optimize the IB tradeoff between minimizing the complexity of their lexicon and maximizing its accuracy. Assuming the listener's inferences are adapted to the speaker, the lexicon is defined by an encoder $q(w|m)$. Complexity in IB roughly corresponds to the number of bits required for communication, formally defined by $I_q(M; W)$, the mutual information between speaker's meanings and words. Accuracy is defined by $I_q(W; U)$, the information that the speaker's words maintain about the target world state, which is also inversely related to the KL-divergence between the speaker's mental state and the listener's inferred state, $\mathbb{E}_q[D[M\|\hat{M}]]$ (Harremoes & Tishby, 2007; Zaslavsky, 2020). An optimal lexicon, or semantic system, is one that minimizes the IB objective function

$$\mathcal{F}_\beta[q] = I_q(M; W) - \beta I_q(W; U), \tag{1}$$

where $\beta \geq 0$ controls the complexity-accuracy tradeoff. The solutions to this optimization problem define the IB theoretical limit of efficiency.

This framework generates precise quantitative predictions that have been gaining converging empirical support across hundreds of languages and multiple semantic domains, ranging from perceptually-

grounded domains such as color (Zaslavsky et al., 2018; 2022) to higher-level conceptual domains such as household objects (Zaslavsky et al., 2019; Taliaferro et al., 2025) and personal pronouns (Zaslavsky et al., 2021). In addition, it has been successfully applied to studying emergent communication in artificial agents (Chaabouni et al., 2022; Tucker et al., 2022; Gualdoni et al., 2024; Tucker et al., 2025).

**The IB color naming model.** As part of our evaluation of LLM color naming, we use the previously published IB color naming model from Zaslavsky et al. (2018). The black curve in Figure 3 shows the IB bound for color naming from this study, together with a reproduction of their results showing that color naming systems across languages (from the WCS as well as English from Lindsey & Brown (2014)) achieve near-optimal IB tradeoffs.

## 2.3 ITERATED LANGUAGE LEARNING

The second main theoretical framework we apply in this work is based on iterated learning (IL). IL is a paradigm in cognitive science for simulating cultural transmission and eliciting prior inductive biases (Griffiths & Kalish, 2007; Kirby et al., 2008; Griffiths et al., 2008). In a typical IL experiment, participants form chains of "generations." At each generation $t$, a participant is exposed to data $d_{t-1}$ from the previous generation and then produces responses that become input for the next generation. In iterated language learning (ILL), participants learn examples of pairs of stimuli and artificial labels during a training period, after which they assign labels to new, unlabeled stimuli in the same meaning space. In doing so, participants produce a full category system $L_t$, which is sampled from to provide training examples to the next generation, and the process repeats. This process, which requires generalization from limited data, reveals learners' inductive biases for certain linguistic or category structures (Griffiths & Kalish, 2007; Griffiths et al., 2008). As shown in Griffiths & Kalish (2007), under certain conditions, namely that the IL agents are Bayesian who share priors and likelihood functions, this Markov chain converges to a stationary distribution over languages equal to the learners' prior distribution $p(L)$. This makes the strong prediction that languages emerging from IL reflect the population's underlying inductive biases. Although this is an asymptotic characterization of IL dynamics, in behavioral experiments with people, researchers observe rapid convergence to highly non-uniform distributions of languages (Kirby et al., 2015).

**Related work on ILL.** Previous research in machine learning has investigated dynamics related to ILL. In agent-based simulations, Ren et al. (2020) introduced the neural iterated learning (NIL) framework, and showed how it can lead to compositional language in neural network agents, and Carlsson et al. (2024) showed that introducing communication-based training in NIL leads to IB-efficient color naming systems. These studies, however, did not explore LLMs as we do here. Zhu & Griffiths (2024) adapted IL to LLMs with strong in-context learning capacities as a prompt-based workflow known as Iterated In-Context Learning (I-ICL) to elicit LLMs' implicit prior distributions over aspects of world knowledge. I-ICL has recently been used to analyze cultural evolution in LLMs (Ren et al., 2024) and to compare visual and linguistic abstractions between LLMs and humans (Kumar et al., 2024). Here, we introduce iterated in-context *language* learning (IICLL, Figure 1c), which goes beyond prior work in leveraging the strong in-context learning abilities of LLMs to replicate as closely as possible the experimental conditions of ILL studies with humans, enabling a direct comparison to LLMs of their respective inductive biases, particularly with regards to semantic efficiency and alignment.

**ILL of color naming systems.** The empirical comparison for our IICLL color naming study is the IL data from Xu et al. (2013). In their experiment, participants were asked to learn and transmit novel systems of color terms across thirteen generations. We focus on their main results that include twenty iterated learning chains, each initialized with a random partition of the WCS grid. These chains vary in the number of allowed color terms, ranging from two to six, and four replications of each condition. Participants were shown a set of randomly selected colors generated uniquely for each chain, and paired with corresponding pseudo words. After training, participants were asked to label all 330 colors of the WCS grid. Xu et al. (2013) found that over time, the IL chains become increasingly regular and resemble the color naming systems documented in the WCS dataset. More recently, Imel et al. (2025) found that these chains also converge to highly efficient systems along the IB bound. Figure 3 shows our reproduction of this finding (plotting only final generations).

# 3 EXPERIMENTAL SETUP

Our goal is to test whether LLMs have an inductive bias toward IB-efficiency, as observed in humans. To this end, we conduct two studies with LLMs: (1) an English color naming study to assess their semantic alignment and communicative efficiency with respect to English speakers; and (2) a cultural transmission experiment of artificial color naming systems (using IICLL) to elicit their inductive learning biases beyond patterns they may have seen during training.

**Models.** We consider 39 models across 6 model families: Gemini (Google, 2025), Gemma 3 (Gemma-Team et al., 2025), Llama 3 (Grattafiori et al., 2024), Qwen 2.5 (Qwen et al., 2025), Olmo 2 (OLMo-Team et al., 2025) and GPT-2 (Radford et al., 2019). Within each family, we vary models along several dimensions. Specifically, to gain insight into the properties that may influence the models' behavior in our tasks, we consider models with different sizes, instruction-tuned versus base models, and text-based versus multi-modal models. For Olmo, we also considered its learning dynamics by analyzing training checkpoints. For a full list of models, see Table 1 in Appendix D.

**Prompts.** In both of our studies, we provided instructions in the prompts to choose only from a fixed set of terms. The Gemini API supports controlled generation which makes this constrained classification task straightforward; for all open-weight models, we used log probability based scoring of the allowed terms as a continuation of the prompt. Further details and example prompts can be found in Appendix J.

**Stimuli.** We used the 330 color chips from the WCS grid shown in Figure 1a. These chips represent a systematic sampling of color space and are standard stimuli in color naming research. Each chip is associated with precise coordinates in the CIELAB color space, which can be converted to sRGB coordinates. To present the color stimuli to the text-based LLMs, we encoded color using these numerical coordinates. For multimodal models, we generated a square colored image corresponding to the WCS chip's sRGB values, and passed this image together with the text instructions.

**Evaluation.** Following Zaslavsky et al. (2018), we use two main evaluation measures in our studies. First, the ***efficiency loss*** of a semantic system is measured by its minimum deviation from optimality, defined as $\varepsilon = \min_\beta \{\frac{1}{\beta}(\mathcal{F}_\beta[q] - \mathcal{F}_\beta^*)\}$ where $\mathcal{F}_\beta^*$ is the optimal value of $\mathcal{F}_\beta$ in Eq. 1. Second, we measure the ***semantic (mis)alignment*** between two systems by the Normalized Information Distance (NID) (Kraskov et al., 2005; Vinh et al., 2010). NID is a metric capturing the distance between two clusterings (in this case, induced by color naming categories), providing a quantitative measure of structural similarity between the naming systems. Since NID is bounded in $[0, 1]$, we take 1 - NID as a measure of similarity, or alignment. ***IB-alignment*** measures the similarity between a system and the nearest ($\epsilon$-fitted) optimal IB system. ***WCS-alignment*** measures the average alignment between a system and the WCS languages, and ***English-alignment*** measures the alignment between a system and English.

# 4 RESULTS

## 4.1 ENGLISH COLOR NAMING

We begin with the results from our English color naming study, in which we elicited color naming responses from LLMs with English color terms and then evaluate their IB-efficiency and alignment with the actual English color naming system from Lindsey & Brown (2014). As illustrated in Figure 1a, we consider two variants of this task, a text-only variant where colors are presented as sRGB coordinates, and an image-based variant where colors are presented as a color patch. More details on our procedure can be found in Appendix B.

The resulting systems are shown in Figure 2b, their IB tradeoffs are shown in Figure 2a, and their quantitative evaluation is shown in Figure 2c (see also Figures 7, 8 and 9 in Appendix E). We find that LLMs vary widely in their complexity and English-alignment, with larger instruction-tuned models achieving better alignment and IB-efficiency. No model aligns perfectly with the English system from Lindsey & Brown (2014), but Gemini-2.0 and Gemma 3 27B (inst.) approximate it closely. The fact that so many state-of-the-art pretrained LLMs struggle to recapitulate the English

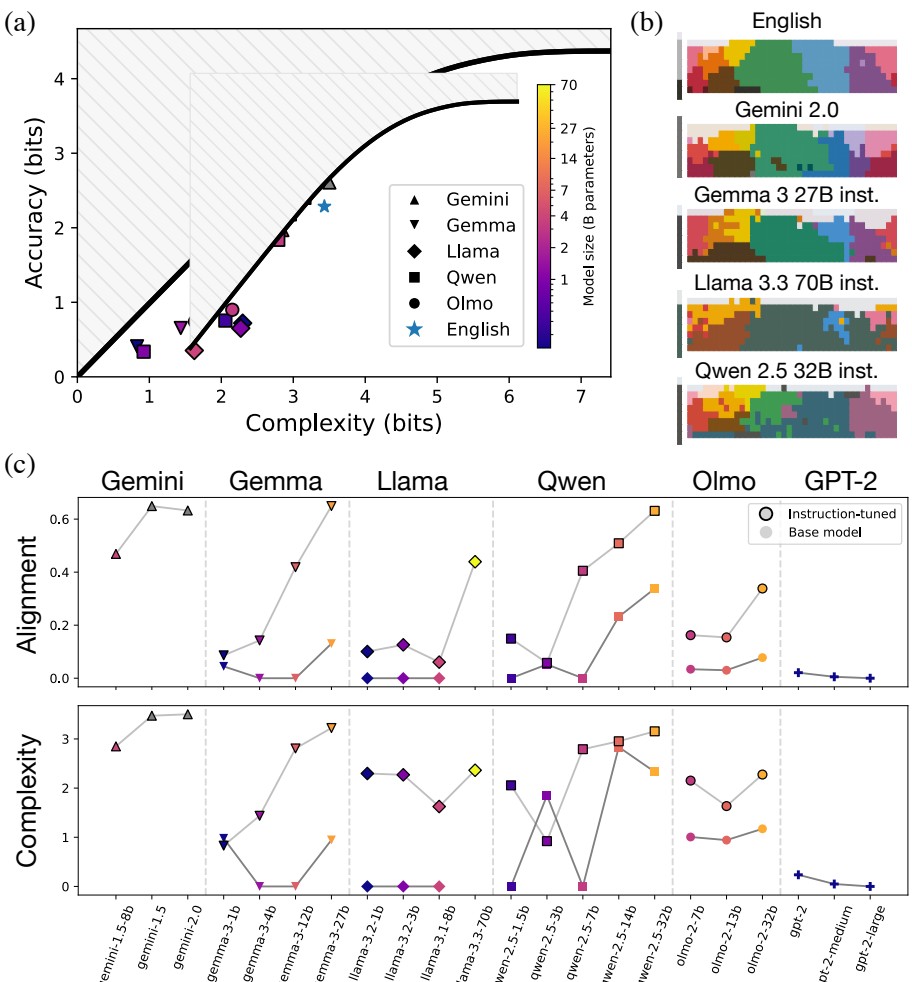

Figure 2: **English color naming experiment with LLMs.** **(a)** IB complexity-accuracy tradeoffs achieved by instruction-tuned LLMs (see Appendix E for base models too), plotted in comparison to the English tradeoff (blue star) and the IB bound (black curve) from Zaslavsky et al. (2018). Models vary widely in their tradeoffs, with larger models approaching the English point. **(b)** Color naming systems of English (Lindsey & Brown, 2014) and best-performing LLMs. Each system is shown by its mode map, i.e., it is plotted against the WCS grid (Figure 1a), where each chip is colored by the color-centroid of its modal category. **(c)** English-alignment (top) and IB complexity (bottom) of all LLMs. Markers are the same as in (a), where a black edge indicates the instruction-tuned model and no edge indicates the base model. Across model families, size and instruction-tuning are associated with higher complexity and better alignment to English.

color naming system, even though these models are trained on massive amounts of English data and have billions of parameters, is quite striking. While instruction tuning is generally associated with better performance, it does not guarantee IB-efficiency or alignment to English, even for very large models (for example, Llama 3.3 70B inst.). At the same time, we were surprised to find that some models—particularly Olmo 2 32B (inst.) and Qwen 2.5 VL 7B (inst.)—produced systems with category structure resembling not English, but instead other, very low-resource languages from the WCS (see Figure 9 in Appendix E). This suggests that although many LLMs struggle to recover the same particular distinctions as English speakers, they may still possess human-like color categories.

To better understand how LLMs acquire color categories during training, we analyzed the learning trajectory of Olmo 2 32B (Appendix F). English-alignment only slightly increases during pre-training, and the most substantial improvement occurs in the second, instruction-tuning stage. This

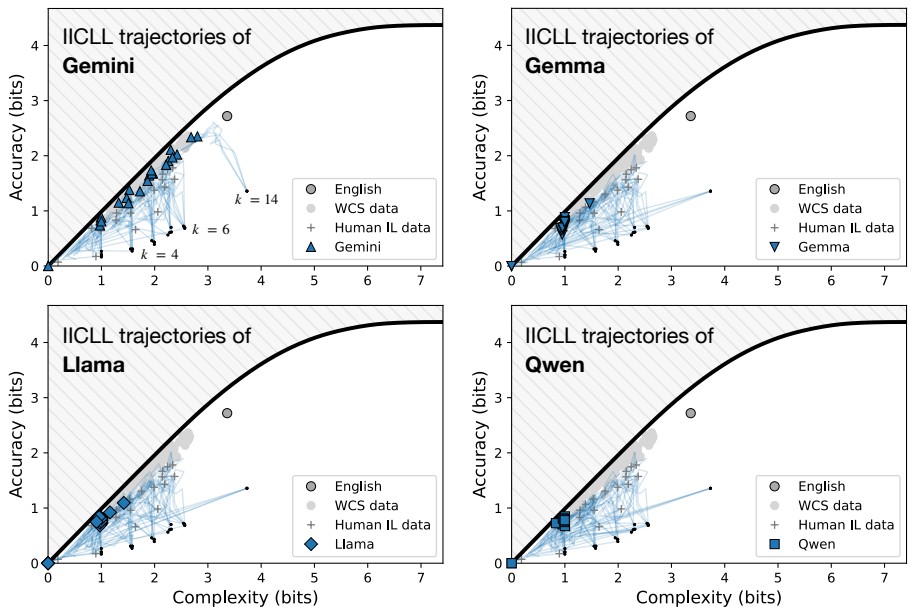

Figure 3: **IICLL with LLMs converges to near-optimal IB solutions.** The trajectories of Gemini 2.0 (upper left), Gemma 3 27B (upper right), Llama 3.3 70B (lower left) and Qwen 2.5 32B (lower right) are plotted on the information plane (same as Figure 2A), together with the IB tradeoffs across human languages (WCS+English) and human IL data. Small black dots correspond to random initializations of chains with varying number of categories, $k \in \{2, 3, 4, 5, 6, 14\}$. Thin blue lines correspond to the LLMs' IICLL trajectories. Gemini captures the complexity range observed across human languages, while the other models converge to lower complexity systems. All models are instruction-tuned.

finding is consistent with the results of Figure 2c, suggesting that instruction-tuning is an important factor in the models' ability to exhibit English-like color naming systems.

Finally, to explore the impact of the input representation, we ran two additional analyses. First, we conducted a minimal pair analysis of the multimodal LLMs (Figure 8, Appendix E), comparing systems that were generated with textual vs. image-based representations of color. Interestingly, presenting colors as images rather than textual sRGB coordinates does not improve the alignment or efficiency of larger models, and can even harm their performance, but it does seem to boost the performance of smaller models. Second, we tested the impact of using CIELAB coordinates instead of sRGB, which better captures human perceptual similarities between nearby colors. Consistent with previous findings by Marjieh et al. (2024), we find that all models, including the best performing ones, struggled to align with English naming when colors are presented in CIELAB. This reveals a key difference between how LLMs represent color and how humans do.

Taken together, our findings show that while many LLMs struggle to align with humans on a very simple, but perceptually-grounded, color-naming task, frontier models do exhibit human-aligned color naming systems. This, in turn, opens a crucial question: is this behavior merely a reflection of imitating patterns in the models' training data, or does it signify a more intrinsic LLM inductive bias towards IB-efficiency in categorization?

## 4.2 ITERATED IN-CONTEXT LANGUAGE LEARNING (IICLL)

To investigate whether LLMs possess an efficiency bias that extends beyond learning the specific categories of a language on which they were trained, we turn to our second study, which simulates cultural transmission in LLMs. To this end, we sought to replicate the ILL color naming experiment of Xu et al. (2013) as closely as possible, using our IICLL paradigm shown in Figure 1b (see Appendix G for more details). We considered only large, instruction tuned models that performed well in the English color naming task for our IICLL experiments: Gemini 2.0, Gemma 3 27B, Qwen

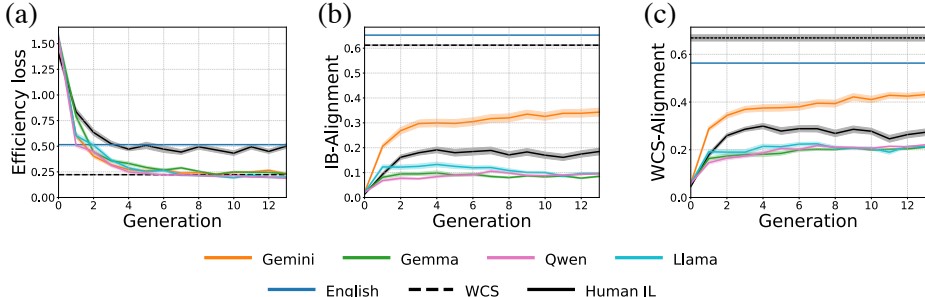

Figure 4: Across IICLL generations, emergent LLM systems become more efficient (a), more aligned with the optimal IB systems (b), and more aligned with human languages (c). Colored curves show the average across initializations and conditions, and the colored regions corresponds to the 95% confidence intervals.

2.5 32B, and Llama 3.3 70B—henceforth, Gemini, Gemma, Qwen and Llama (see Appendix L for an analysis showing that smaller models struggle in IICLL to produce non-degenerate category systems).

Figure 3 shows the resulting trajectories of IICLL chains on the information plane. Gemini develops color naming systems that converge to a similar range of near-optimal IB solutions as the typological patterns of the WCS languages, as well as the final generation systems from the human IL chains from Xu et al. (2013). There is also broad qualitative fit between its final generation IICLL systems and languages from the WCS (see Appendix I). The other LLMs also develop systems that converge to highly IB-efficient solutions, although they appear to be limited to the lower range of complexity observed in the WCS.

Additional quantitative support for these observations is provided in Figure 4. These figures show that over generations, LLM systems become more efficient in their mapping of stimuli to terms (Figure 4a), more similar to naturally occurring human color systems (Figure 4c), and more similar to optimal IB systems (Figure 4b). Furthermore, LLM IICLL chains converge near the bound relatively quickly (after roughly four generations), parallel to human IL dynamics.

Strikingly, many trajectories from all models initially climb in complexity towards the IB bound before slowly evolving downwards alongside it. This suggests that the capacity to learn and transmit complex yet near-optimally efficient category systems, while strongest for Gemini, is present in all four LLMs. One factor that may drive the difference between Gemini and the other models is that the IICLL task requires very strong in-context learning, as models must integrate dozens of in-context training examples to generalize well. For example, the $k = 14$ condition includes $84$ examples, and in this setting most of the LLMs immediately converge to low-complexity solutions. However, given that the terms in our IICLL experiment are made up (pseudo) terms and that we give no indication to the model that the stimuli are in fact colors, only that they have "features" (Figure 1c), all four models show an impressive ability to evolve IB-efficient systems.

To assess whether the emergent LLM systems are non-trivially efficient and aligned to WCS languages, we conducted a rotation analysis (Regier et al., 2007) on the final, evolved color systems from our LLM experiments alongside the human data from Xu et al. (2013) (Figure 11 in Appendix H). This analysis involved rotating the color-label mapping along the hue dimension of the WCS color grid (i.e., along the columns of Figure 1a) and evaluating the difference in their efficiency and alignment scores. We find that such rotations away from the actual emergent systems lead to a significant decrease in efficiency and alignment for Gemini, while the results are less conclusive for the other models. Furthermore, we find that Gemini's efficiency and alignment over generations are higher than the human IL trajectories, and are also higher than a baseline learner based on an alternative feature-based clustering algorithm (Appendix M). This suggests that Gemini truly exhibits an emergent inductive learning bias toward IB-efficiency, even though as fas as we know it was not trained or fine-tuned with respect to the IB objective function.

(a) 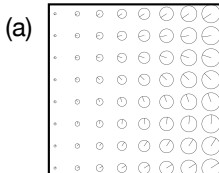   (b) 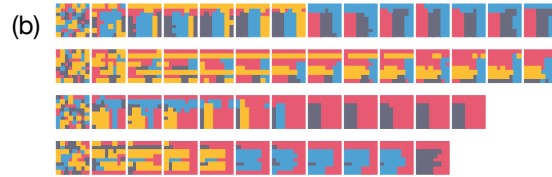

Figure 5: (a) The Shepard circles stimulus grid, (b) Gemini IICLL chains for naming Shepard circles. Rows correspond to individual chains, initialized randomly. Each system is plotted over the stimulus grid, where colors correspond to unique labels.

### 4.3 THE EMERGENCE OF CATEGORIES IN SHEPARD CIRCLES VIA IICLL

While many results in color have previously been shown to generalize to other domains (Zaslavsky et al., 2019; 2021; Mollica et al., 2021), it is challenging to conduct in other domains a full-scale analysis of the scope we performed for color naming, primarily due to the lack of high-quality data from both native speakers and ILL experiments. Here, nevertheless, we apply the IICLL paradigm in a qualitatively distinct semantic domain to provide initial evidence that our results may indeed generalize beyond color.

To this end, we considered a synthetic domain of so-called "Shepard circles," a classic conceptual space used to study how humans categorize multidimensional stimuli (Shepard, 1964). These are circles which vary in both radius and angle of rotation of an internal spoke. Following Carr et al. (2020), we generated stimuli by taking eight evenly spaced values for each of the two dimensions, yielding $64$ total stimuli (Figure 5a). For this preliminary investigation, we limit our analysis to Gemini and $k = 4$ allowed labels. We found that presenting these stimuli as pairs of numbers (radius and angle) proved to be challenging. This is perhaps unsurprising, because unlike color—which is frequently represented in various ways online in both text and images—there is likely no text online that would allow the model to associate these numbers with meaningful perceptual features. To overcome this limitation and better mimic human perception, we presented Gemini with images of the stimuli. Four samples of IICLL chains are are shown in Figure 5b. Over generations, Gemini transmitted categories that became increasingly compact in their partitioning of the space, and distinguished regions based on both the radius and angle of the circles. This suggests that LLMs—especially frontier multimodal models—potentially have a domain-general bias to organize features into non-arbitrary, and increasingly regular, semantic categories. An important direction for future work is to test whether this emergent structure also supports greater IB-efficiency as seen in humans (Imel et al., 2025).

## 5 DISCUSSION

In this work, we combined a theory-driven approach, based on the IB principle, with cognitively-motivated experimental methods, based on color naming and iterated language learning to study whether LLMs can acquire a human-like inductive bias toward optimally-compressed semantic representations, without being trained for this objective. To do this, we first conducted an in-depth analysis of English color naming across 39 LLMs, and found that a surprising number of state-of-the-art models fail to capture the English color naming system. However, some of the most recent, larger, instruction-tuned models achieve high English-alignment and comparable IB tradeoffs. We then demonstrated that LLMs that do align well with the English color naming system are not merely mimicking patterns in their training data, but rather exhibit a more fundamental capacity to learn human-aligned, efficient color category systems. To do this, we introduced Iterated in-Context Language Learning (IICLL) to simulate cultural transmission of category systems. Over generations of IICLL, LLMs tend to restructure randomly-initialized artificial category systems toward greater IB-efficiency and alignment to human naming systems. We also provide initial evidence that LLMs can develop structured categories over generations of IICLL in a domain distinct from color, suggesting that our results may apply in other semantic domains as well. Taken together, our findings suggest

that LLMs are capable of evolving perceptually grounded, human-like semantic systems, guided by the same IB-efficiency principle that underlies human languages. Importantly, neither humans nor LLMs are explicitly trained for optimizing the IB objective, suggesting that IB-efficiency may emerge to support intelligent behavior.

Our empirical results open up several important questions for future research. First, while our work demonstrates that cultural transmission alone (via IICLL) can be a sufficient pressure for some LLMs to develop efficient, human-like category systems, a more complete understanding of language evolution requires integrating functional pressure of language use, e.g., via communication. Therefore, an important future direction is to extend our IICLL framework to incorporate communication as a selective pressure, for example, by adopting models that explicitly integrate both transmission and communication (e.g., Kouwenhoven et al., 2024). Second, the precise origins of the bias we observe in LLMs toward efficiency are unclear (for example, how might this bias emerge from properties of the training data, instruction-tuning, or model size), and investigating this is another important direction for future work. Lastly, it is important to extend our analyses across more languages and semantic domains.

## REPRODUCIBILITY STATEMENT.

This work utilizes a previously published model and code from Zaslavsky et al. (2018), made available at `https://github.com/nogazs/ib-color-naming` under the MIT License, and code from Carr et al. (2020), made available from `https://github.com/jwcarr/shepard` under a CC BY 4.0 License. The ILL data from Xu et al. (2013) were obtained from the authors, and the WCS data are publicly available from the World Color Survey website, `https://linguistics.berkeley.edu/wcs`. Appendix D provides a list of the specific models used with their Hugging Face or Google API IDs, and Appendix J includes example prompts. Code for reproducing our experiments can be found at `https://infocoglab.github.io/evolution-compression-llms`. To support full reproducibility, the LLM data that was generated for this paper is available upon request.

## ACKNOWLEDGMENTS

This work was supported by a Google Cloud Platform Credit Award and a BSF Research Grant (2024318) to NZ. An earlier version of this work received the best paper award at the NeurIPS 2025 Workshop on Interpreting Cognition in Deep Learning Models (CogInterp).

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

## A    IB COMMUNICATION MODEL AND THEORETICAL BOUND

**IB communication model**    The IB framework is based on a basic communication model that considers an inventory of words $\mathcal{W}$ to communicate about a space of meanings, $\mathcal{M}$. The specific model we apply in this paper is the previously published IB color naming model of Zaslavsky et al. (2018). In this model, meanings are assumed to be mental representations or beliefs over world states $\mathcal{U}$, formally defined as probability distributions $m(u)$ over world states $u \in \mathcal{U}$. In our setting, the set of world states $\mathcal{U}$ is taken to be the 330 color chips from the WCS grid, and meanings over colors are grounded in the CIELAB perceptual space such that each target color referent, $u_t \in \mathcal{U}$, is represented as a Gaussian distribution $m_t(u)$ centered around $u_t$. Meanings are drawn from an information source, $p(m)$, which characterizes how often each meaning needs to be communicated. The need distribution over meanings in this model, $p(m)$, was estimated using the method of least-informative priors (see Zaslavsky et al. (2020) for an extensive evaluation of this need distribution). Note that, in our IICLL experiments, we sample stimulus-word pairs uniformly for training data for each generation, instead of biased sampling from this prior. This was done in order to replicate the procedure in Xu et al. (2013).

Given a meaning $m \sim p(m)$, a speaker produces a signal $w$ using a stochastic production policy, also called an encoder, $q(w|m)$, and then a listener interprets the signal by reconstructing an estimated belief state $\hat{m}_w(u) = \sum_m q(m|w)m(u)$. Note that this form of listener interpretations corresponds to a Bayesian listener. While this form is assumed here for simplicity, it is not an actual assumption but rather a derivation from the theory (see the SI of Zaslavsky et al. (2018)).

**IB theoretical bound**    A semantic category system in this framework corresponds to a stochastic encoder, $q(w|m)$, which maps meanings to signals. An optimal semantic system, according to the IB principle, is an encoder that satisfies a tradeoff between its informational complexity and communicative accuracy. Complexity, also known as information rate (Cover & Thomas, 2006), is defined by the mutual information between speaker meanings and signals,

$$I_q(M; W) = \sum_{m,w} p(m)q(w|m) \log \frac{q(w|m)}{q(w)}, \tag{2}$$

which captures the number of bits, on average, that is required to encode meanings with signals. Maintaining low complexity corresponds to using fewer bits for communication, i.e., achieving high compression rate, which in turn, can be translated to affording a smaller lexicon size. For example, minimal complexity can be achieved by compressing all possible meanings into a single signal. This, however, will result in very poor accuracy. Accuracy, in IB terms, is quantified by

$$I_q(W; U) = I(M; U) - \mathbb{E}_q \left[ D \left[ M \| \hat{M} \right] \right], \tag{3}$$

which corresponds to maintaining a language that is informative about the speaker's intentions. Maximizing accuracy, as expressed in Eq. 3, amounts to minimizing the expected distortion between speaker and listener meanings, i.e., the Kullback-Leibler (KL) divergence $D[M \| \hat{M}]$. We say that a language's semantic system is *efficient* to the extent that its encoder $q$ minimizes the IB tradeoff:

$$\mathcal{F}_\beta[q] = I_q(M; W) - \beta I_q(W; U) \text{ s.t. } \beta \geq 1, \tag{4}$$

where $\beta$ is a free parameter controlling the tradeoff between pressure to minimize complexity and pressure to maximize accuracy. The solutions to this optimization problem define the IB theoretical limit of efficiency, which means that no system can lie above this theoretical bound (see Fig. 3).

## B ENGLISH COLOR NAMING PROCEDURE

The color naming task was conducted by prompting a model to label a color chip, for all 330 color chips from the WCS stimulus array, one stimulus at a time, in a randomized order. Unlike the prompts in our IICLL study, we do not provide the model with access to previous interactions. Each prompt was of the form: "What color is this [ `<color coordinates>` ]? You may only use one of the following allowed labels: ['Red', 'Blue', . . . ] Please provide only a single label from the list just provided. Do not give any explanation." When prompting models with text only, the color coordinates were sRGB triples. The allowed set of labels were fourteen basic English color terms corresponding to the modal terms from Lindsey & Brown (2014).

## C ENGLISH COLOR NAMING WITH CIELAB INPUTS

To address the generalizability of our findings beyond sRGB inputs, we replicated our English color naming task using CIELAB triples as the input features for the prompt text. CIELAB is a perceptually uniform space and so provides a principled color space to contrast with sRGB. We restricted to testing the four models that performed best from our original study (which used sRGB-encoded stimuli).

Figure 6 illustrates mode maps of the resulting systems for models when prompted with CIELAB-encoded colors. We found that performance generally degraded across all models when providing features as CIELAB triples: they struggled significantly to align with established English color boundaries, and the resulting category systems were considerably noisier compared to the sRGB-based outputs. This outcome is consistent with previous research by Marjieh et al. (2024) which noted that other frontier models failed to produce coherent English color category systems when prompted with CIELAB triples. Based on this observation, we restricted our IICLL study to the sRGB feature space to assess the models' inductive bias under ideal conditions.

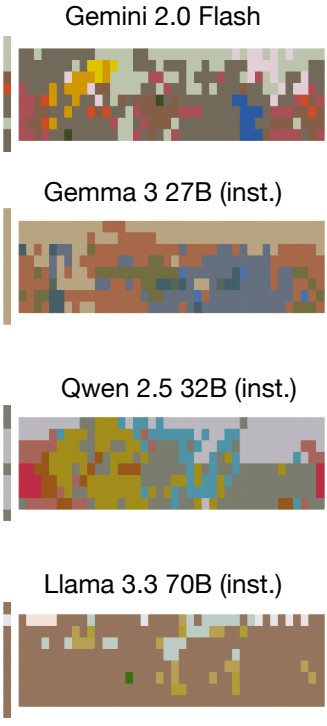

Figure 6: Mode maps for the best-performing models from our English color naming study, when prompted with CIELAB triples.

# D  MODELS

| Model | Size (B params) | Instruction Tuned | Multimodal | Hugging Face / Google API ID |
|---|---|---|---|---|
| gemini-1.5 | n/a | Yes | Yes | gemini-1.5-flash |
| gemini-1.5-8b | 8 | Yes | Yes | gemini-1.5-flash-8b |
| **gemini-2.0** | n/a | Yes | Yes | gemini-2.0-flash |
| gemma-3-1b | 1 | No | No | google/gemma-3-1b |
| gemma-3-1b-it | 1 | Yes | No | google/gemma-3-1b-it |
| gemma-3-4b | 4 | No | Yes | google/gemma-3-4b |
| gemma-3-4b-it | 4 | Yes | Yes | google/gemma-3-4b-it |
| gemma-3-12b | 12 | No | Yes | google/gemma-3-12b |
| gemma-3-12b-it | 12 | Yes | Yes | google/gemma-3-12b-it |
| gemma-3-27b | 27 | No | Yes | google/gemma-3-27b |
| **gemma-3-27b-it** | 27 | Yes | Yes | google/gemma-3-27b-it |
| gpt-2 | 0.224 | No | No | openai-community/gpt2 |
| gpt-2-medium | 0.355 | No | No | openai-community/gpt2-medium |
| gpt-2-large | 0.774 | No | No | openai-community/gpt2-large |
| llama-3.1-8b | 8 | No | No | meta-llama/Llama-3.1-8B |
| llama-3.1-8b-instruct | 8 | Yes | No | meta-llama/Llama-3.1-8B-Instruct |
| llama-3.2-1b | 1 | No | No | meta-llama/Llama-3.2-1B |
| llama-3.2-1b-instruct | 1 | Yes | No | meta-llama/Llama-3.2-1B-Instruct |
| llama-3.2-3b | 3 | No | No | meta-llama/Llama-3.2-3B |
| llama-3.2-3b-instruct | 3 | Yes | No | meta-llama/Llama-3.2-3B-Instruct |
| **llama-3.3-70b-instruct** | 70 | Yes | No | metallama/Llama-3.3-70B-Instruct |
| olmo-2-7b | 7 | No | No | allenai/OLMo-2-7B |
| olmo-2-7b-instruct | 7 | Yes | No | allenai/OLMo-2-7B-Instruct |
| olmo-2-13b | 13 | No | No | allenai/OLMo-2-13B |
| olmo-2-13b-instruct | 13 | Yes | No | allenai/OLMo-2-13B-Instruct |
| olmo-2-32b | 32 | No | No | allenai/OLMo-2-32B |
| olmo-2-32b-instruct | 32 | Yes | No | allenai/OLMo-2-32B-Instruct |
| qwen-2.5-1.5b | 2 | No | No | Qwen/Qwen-2.5-1.5B |
| qwen-2.5-1.5b-instruct | 2 | Yes | No | Qwen/Qwen-2.5-1.5B-Instruct |
| qwen-2.5-3b | 3 | No | No | Qwen/Qwen-2.5-3B |
| qwen-2.5-3b-instruct | 3 | Yes | No | Qwen/Qwen-2.5-3B-Instruct |
| qwen-2.5-7b | 7 | No | No | Qwen/Qwen-2.5-7B |
| qwen-2.5-7b-instruct | 7 | Yes | No | Qwen/Qwen-2.5-7B-Instruct |
| qwen-2.5-vl-7b-instruct | 7 | Yes | Yes | Qwen/Qwen-2.5-VL-7B-Instruct |
| qwen-2.5-vl-32b-instruct | 32 | Yes | Yes | Qwen/Qwen-2.5-VL-32B-Instruct |
| qwen-2.5-14b | 14 | No | No | Qwen/Qwen-2.5-14B |
| qwen-2.5-14b-instruct | 14 | Yes | No | Qwen/Qwen-2.5-14B-Instruct |
| qwen-2.5-32b | 32 | No | No | Qwen/Qwen-2.5-32B |
| **qwen-2.5-32b-instruct** | 32 | Yes | No | Qwen/Qwen-2.5-32B-Instruct |

Table 1: Full list of models used in our naming study. Models used in our IICLL task are bolded.

# E  NAMING SYSTEMS FOR ALL MODELS

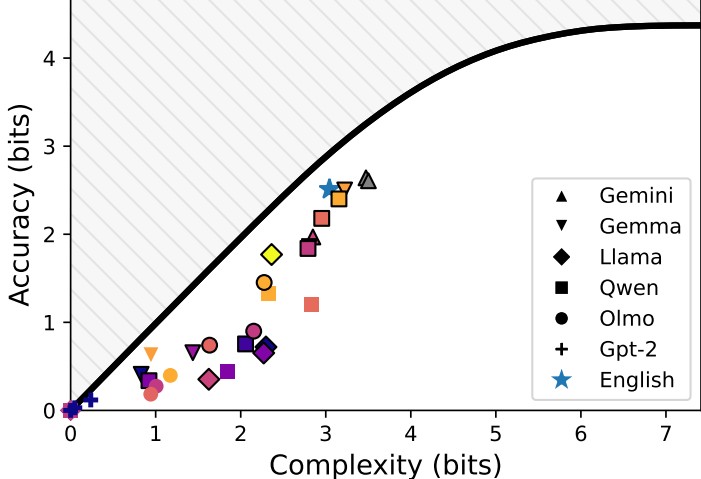

Figure 7: IB complexity-accuracy tradeoffs achieved by text-based LLMs. Same as Figure 2a but including base (not instruction-tuned) models. Points with a black edge indicate models that are instruction-tuned.

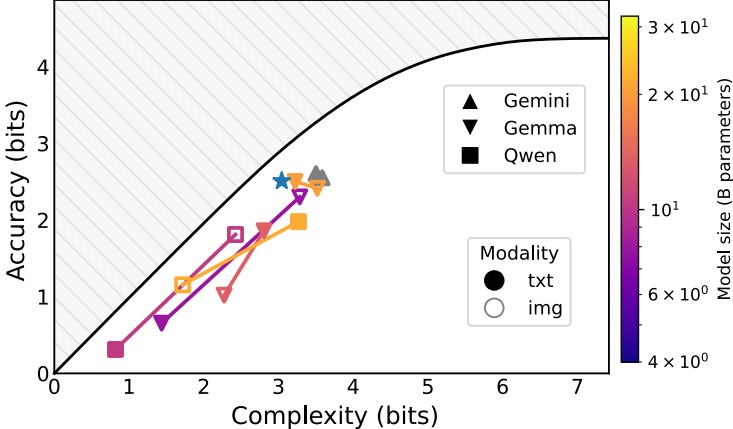

Figure 8: IB complexity-accuracy tradeoffs achieved by instruction-tuned multimodal LLMs. Points without color filling indicate systems resulting from providing colors as images, rather than via sRGB coordinates in text. Each minimal pair is connected by a line. Interestingly, there appears to be a cutoff at around 3 bits of complexity for which models that already perform well with text-based prompting do not improve alignment to English when prompted with an image instead.

As can be seen in Figure 9, many of the models performed poorly on the English naming task, with many base (not instruction-tuned) models and some smaller instruction-tuned models failing to produce naming systems with any coherent category structure. Figure 7 shows that these systems are distributed widely across the information plane, with only larger, instruction-tuned, very recent models achieving similar IB-efficiency tradeoffs to English speakers. Interestingly, some models show similar category structure not to English, but to languages in the WCS. For example, Olmo-2-32B-Instruct, and Qwen-2.5-VL-7B-Instruct (when presented with images) bear resemblance to Mayoruna (see the final row of Figure 13), as well as some of the systems to which Gemini-2.0 converges over generations of IICLL. This suggests that although some larger, instruction tuned

# Gemini

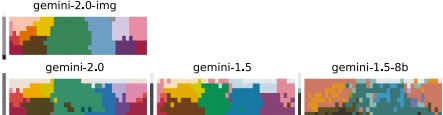

# Gemma

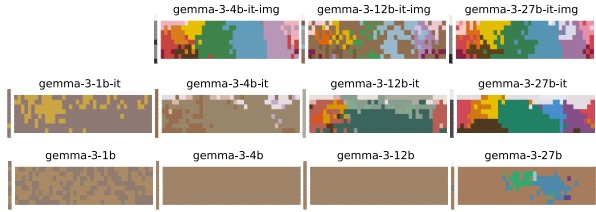

# Llama

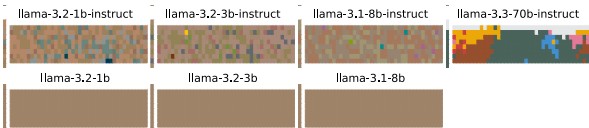

# Qwen

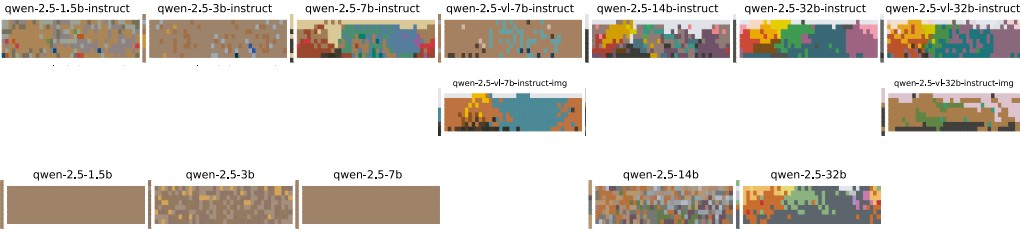

# Olmo

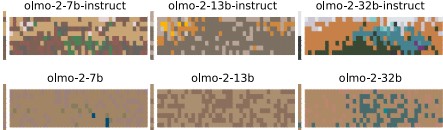

# GPT-2

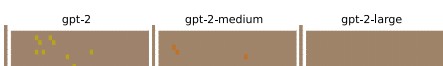

Figure 9: Mode maps for all models. The suffix '-img' for multimodal models indicates that images were presented as the color stimulus, rather than sRGB coordinates.

models and multimodal instruction-tuned models fail to recover the English naming system, they may still possess some general bias towards color category structure that is aligned with humans.

## F OLMO TRAINING TRAJECTORY

As model checkpoints are available for the base Olmo 2 32B (OLMo-Team et al., 2025) model throughout its training, we repeated our English color naming study throughout different stages of its model training. The training procedure involved two main phases: pre-training on up to 6 trillion tokens, and a second training stage using a curated dataset of 843 billion tokens, including high-quality, academic, and instruction-tuning data. This also involved training on 100 billion and 300 billion token samples and then averaging the final checkpoints in a process called model souping.

Furthermore, we found that hinting to the model that the coordinates were in fact sRGB values could improve model performance—specifically, when the prompt was changed to "What color is this rgb=[ `<color coordinates>` ]?". To aid in our analysis of the base pretrained Olmo models, we used this form of prompting in order to see what structure is achievable in principle.

Figure 10 shows that there is a slight increase in scores towards the end of stage 1 training, and a large jump in performance occurs after just 1000 steps of stage 2 training. This is particularly interesting given that stage 2 included *pretraining* on instruction-tuning data, given that we found instruction-tuning was strongly associated with higher alignment and complexity across other models.

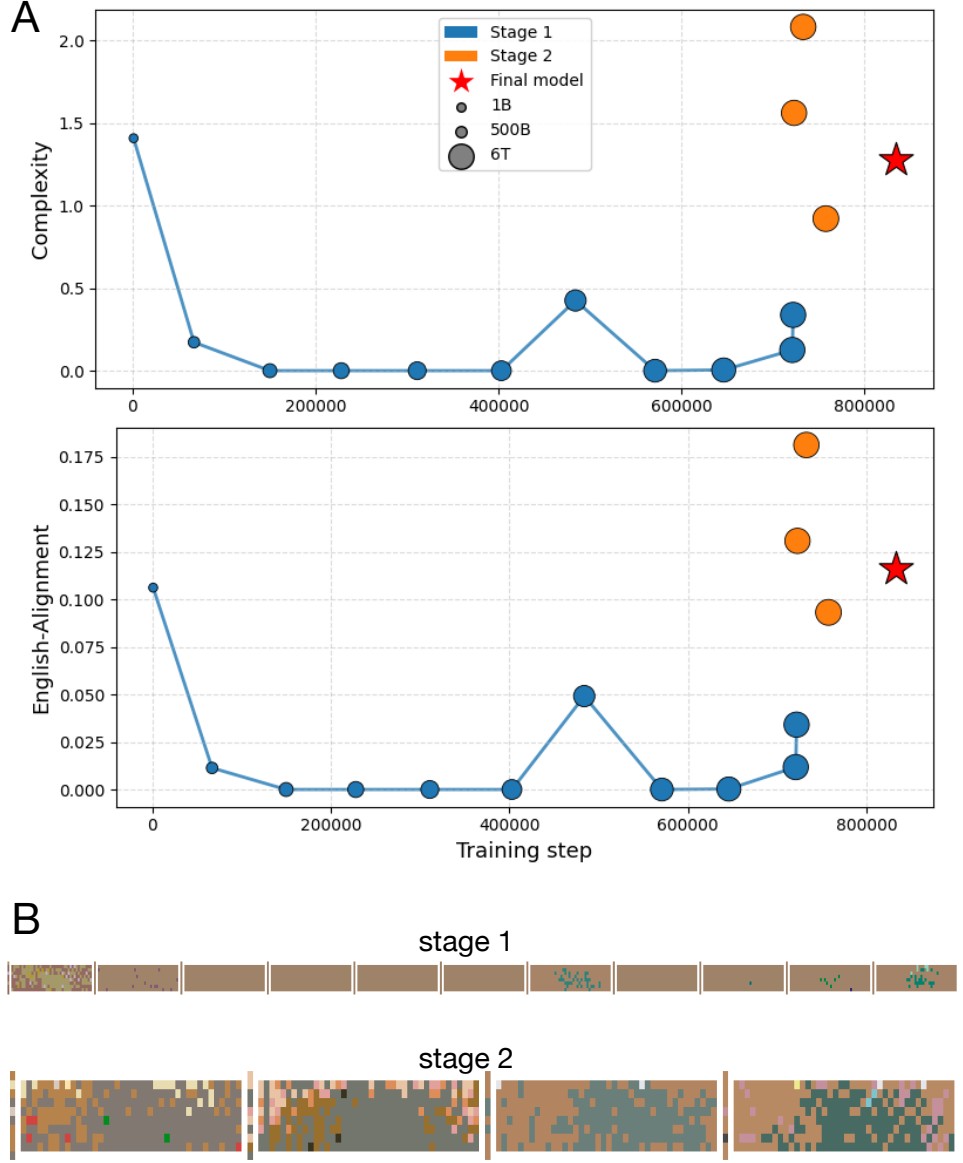

Figure 10: **(a)** Trajectory of English-alignment and complexity of Olmo 2 32B systems over the course of training. **(b)** Mode maps of Olmo systems over the course of training.

# G    Iterated in-context language learning procedure

**Procedure**    We aimed to replicate the procedure of Xu et al. (2013). The initial training data for the first generation of IICLL consisted of random pairings of colors and nonsense terms, with a sample size six times the allowed vocabulary size. For vocabulary sizes ranging from 2 to 6 terms, the initial training data was sourced directly from Xu et al. (2013) to facilitate direct comparison. A 14-term condition was also included for comparison with English (Lindsey & Brown, 2014), with its initial mappings randomly generated as it was not part of the original study.

Next, we performed a sanity check to ensure that the LLM could recover the information from the training data. This involved presenting each stimulus from the training set to the LLM one at a time, in a random order and without any history of previous interactions, and verifying that the model could recover the correct labels as provided in the prompt. With the exception of the most resource-intensive $k = 14$ condition, all models achieved a stable comprehension accuracy of $\sim 80\%$ or greater throughout generations. The $k = 14$ condition served as an extreme-case sanity check, resulting in highly unstable performance (based on a failure to retrieve the required 84 examples; see the final paragraph of this section). Only Gemini occasionally passed this threshold; we elected to report the $k = 14$ condition results to illustrate the point of divergence in model capability.

Following this, we began the 'production phase', in which we prompted the model to provide a label for an unseen color stimulus. This was repeated for every item in the WCS stimulus array, presented in a randomized order, including those stimuli that were part of the initial training set for that generation. During this production phase, we also monitored the LLM's consistency with the labels provided in the initial training set. While perfect adherence was not crucial since deviations may reflect the models' inductive biases, we observed that consistency with the training set remained much higher than chance in the first generations and quickly rose to ceiling.

In addition, to promote response coherence and crudely mimic short-term memory influences found in human experiments, each subsequent prompt was augmented with a sliding window of the 10 most recent user-model interactions (see Appendix K for details on window size selection and its impact).

Once labels were obtained for all stimuli in the production phase, this full set of stimulus-label pairs constituted the current generation's complete color category system. This marked the completion of one generation. To initiate the next generation, a new training set was created by randomly sampling stimulus-word pairs from the just-completed generation's system, with the sample size being six times the number of allowed words in the vocabulary. This entire process was repeated for up to 13 generations, (which was the length of chains in Xu et al. (2013)) or until two successive generations yielded degenerate systems (where all stimuli are mapped to a single term).

# H ROTATION ANALYSIS

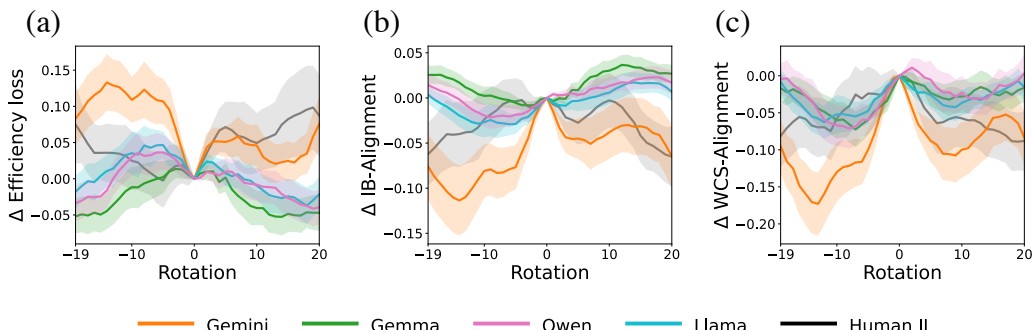

Figure 11: Rotation analysis for final systems of all models' IICLL chains and human ILL chains from Xu et al. (2013). The $x$-axis denotes rotation along columns (hue dimension) of the WCS grid, while the $y$-axis is the difference ($\Delta$) in mean (a) efficiency loss, (b) alignment to IB optima, or (c) alignment to languages in the WCS. Each colored curve is the average across initializations and conditions, and the colored region corresponds to the 95% confidence interval around the mean (assuming a normal distribution). In (a), $\Delta > 0$ means that the actual systems are more efficient that their rotated counterparts, whereas in (b) and (c) $\Delta < 0$ means that the actual systems are more aligned.

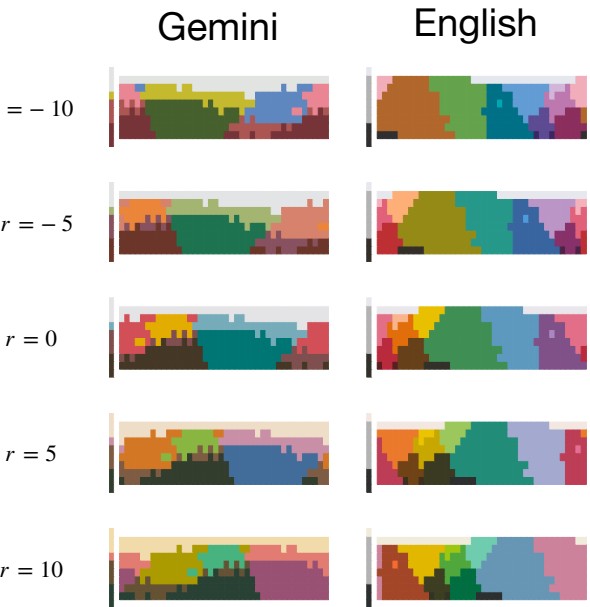

Figure 12: Rotation examples. Hypothetical variants for a final generation system of IICLL with Gemini 2.0 in the $k = 14$ condition, and the English system from Lindsey & Brown (2014). Variants are obtained by rotating the color naming system in the hue dimension across the columns of the WCS stimulus palette. $r = 0$ corresponds to the actual system, $r = 5$ corresponds to a shift of five columns to the right, and $r = -5$ corresponds to a shift of five columns to the left.

# I EXAMPLE CHAINS COMPARED TO WORLD COLOR SURVEY LANGUAGES

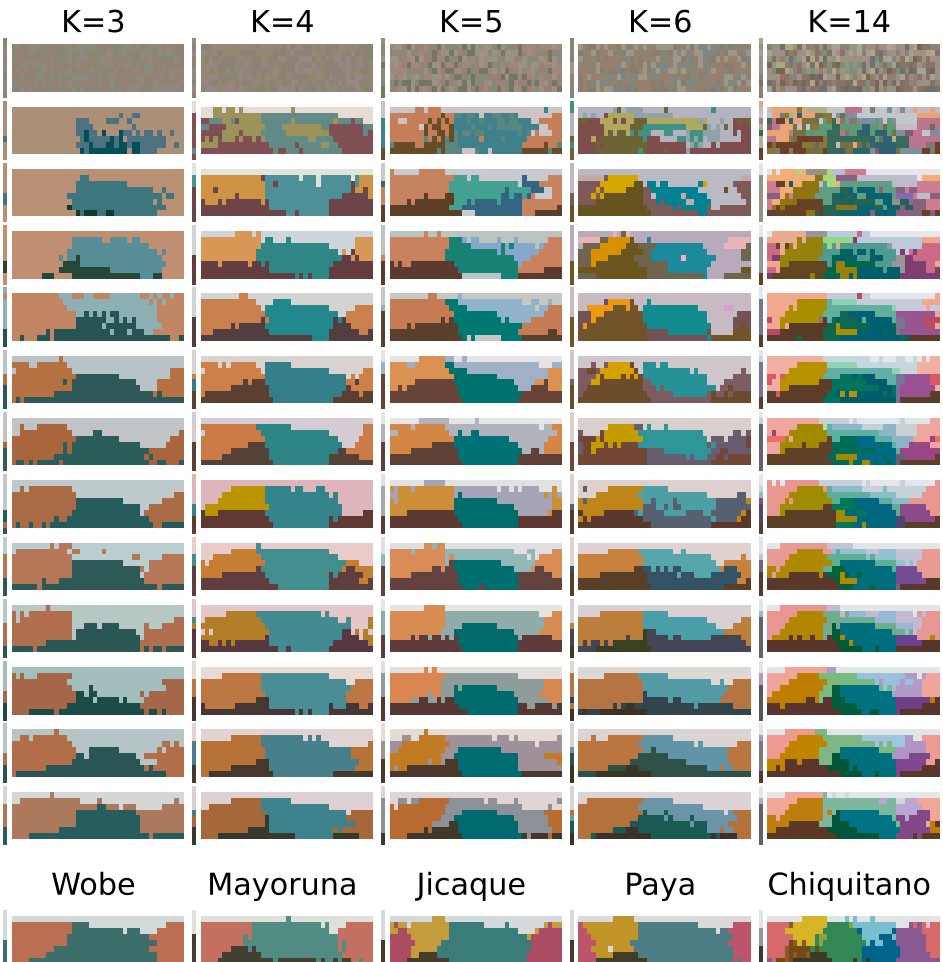

Figure 13: Mode maps for Gemini-2.0 systems over generations of IICLL. Bottom row shows the language from the WCS dataset that is most similar to the final generation's system.

## J    PROMPTING

In both of our studies, we provided instructions in the prompts to choose only from a fixed set of terms. The Gemini API supports controlled generation which makes this constrained classification task straightforward. For all open-weight models, we used log probability based scoring of the allowed terms as a continuation of the prompt to implement constrained choice. We used this decoding approach because after initial exploration of a brute-force method of repeating the prompt up to ten times until the model generated an output from the allowed set of terms, we found that only Llama 3.3-70B-Instruct could generate usable output (and in this case produced comparable behavior). For our probability-based decoding, we used the default generation configurations loaded from the Hugging Face Transformers library, which include a decoding temperature of $0.6$ and a $\texttt{top\_p}$ sampling threshold of $0.9$.

### J.1    EXAMPLE PROMPT FOR ENGLISH COLOR NAMING TASK

```
''What color is this [0.73579176, 0.13100809, 0.20245084]? You may only use one of the
    following allowed labels: ['Red', 'Blue', 'Yellow', 'Green', 'Orange', 'Purple', 'Pink',
    'Brown', 'Black', 'White', 'Gray', 'Peach', 'Lavender', 'Maroon']. Please provide only a
    single label from the list just provided. Do not give any explanation.''
```

### J.2    EXAMPLE PROMPT DURING ITERATED LEARNING

**Training**    During training, the instructions were:

```
[
  {
    "role": "user",
    "content": [
      {
        "type": "text",
        "text": "Features: [0.73579176, 0.13100809, 0.20245084] -> Label: Tovo
                Features: [0.0, 0.32875953, 0.29290289] -> Label: Feglu
                Features: [0.27329472, 0.29777161, 0.12539189] -> Label: Feglu
                Features: [0.18075972, 0.20165954, 0.14769053] -> Label: Narp
                Features: [0.77448248, 0.32302429, 0.52727771] -> Label: Zarn
                Features: [0.49405556, 0.30562009, 0.57919747] -> Label: Mib
                Features: [0.71954112, 0.66114241, 0.82844603] -> Label: Mib
                Features: [0.13380154, 0.21466098, 0.10314594] -> Label: Tovo
                Features: [0.99613596, 0.732415, 0.63294812] -> Label: Tovo
                Features: [0.77102815, 0.83377671, 0.0] -> Label: Blim
                Features: [0.94004023, 0.73594053, 0.83545817] -> Label: Blim
                Features: [0.77780255, 0.4893714, 0.71447577] -> Label: Narp
                Features: [0.89354841, 0.92711068, 0.55762157] -> Label: Zarn
                Features: [0.18901356, 0.19997485, 0.13980956] -> Label: Blim
                Features: [0.26099322, 0.1368575, 0.35608507] -> Label: Zarn
                Features: [0.0, 0.67091016, 0.50450556] -> Label: Mib
                Features: [0.86141792, 0.28265837, 0.0] -> Label: Feglu
                Features: [0.0, 0.42196565, 0.49203637] -> Label: Mib
                Features: [0.80047349, 0.92499868, 0.91404717] -> Label: Mib
                Features: [0.27606817, 0.14437327, 0.29090483] -> Label: Zarn
                Features: [0.91415567, 0.59970537, 0.67293472] -> Label: Narp
                Features: [0.8590007, 0.65998705, 0.0] -> Label: Feglu
                Features: [0.45183228, 0.37479448, 0.08237481] -> Label: Zarn
                Features: [0.91680269, 0.63513813, 0.0] -> Label: Feglu
                Features: [0.8994021, 0.89968419, 0.89916582] -> Label: Blim
                Features: [0.58159027, 0.030559, 0.08701426] -> Label: Blim
                Features: [0.84117237, 0.91203955, 0.93875183] -> Label: Feglu
                Features: [0.12209584, 0.66759353, 0.39283492] -> Label: Tovo
                Features: [0.81026541, 0.67994708, 0.0] -> Label: Blim
                Features: [0.81051796, 0.77667486, 0.88512298] -> Label: Feglu
                Features: [0.47551109, 0.20309021, 0.05617448] -> Label: Mib
                Features: [0.70626646, 0.35443549, 0.64213025] -> Label: Mib
                Features: [0.36974002, 0.20933179, 0.50495342] -> Label: Zarn
                Features: [0.0, 0.44248437, 0.46499573] -> Label: Tovo
                Features: [0.29926292, 0.15509473, 0.11356989] -> Label: Feglu
                Features: [0.0, 0.33817158, 0.21461567] -> Label: Mib"
      }
    ]
  },
  {
    "role": "assistant",
    "content":
```

```
  },
  ...
]
```

**Generalization** During generalization sessions, the prompt included (i) the entire training set, (ii) up to 10 previous user-assistant interaction pairs to serve as a form of memory / history of the conversation, and (iii) the current stimulus being queried from the language model.

For the history, previous interactions appeared in the form:

```
[
  {
    "role": "user",
    "content": [
      {
        "type": "text",
        "text": "Based on the preceding examples, what is the label that best describes this?
            Do not give any explanation, and limit your response to exactly one word from
            this list of labels: ['Narp', 'Tovo', 'Feglu', 'Mib', 'Blim', 'Zarn'].\nFeatures:
            [0.59395637, 0.24607302, 0.5432978] -> Label: "
      }
    ]
  },
  {
    "role": "assistant",
    "content": "Mib"
  },
  {
    "role": "user",
    "content": [
      {
        "type": "text",
        "text": "Based on the preceding examples, what is the label that best describes this?
            Do not give any explanation, and limit your response to exactly one word from
            this list of labels: ['Narp', 'Tovo', 'Feglu', 'Mib', 'Blim', 'Zarn'].\nFeatures:
            [0.73535226, 0.11620602, 0.27974255] -> Label: "
      }
    ]
  },
  {
    "role": "assistant",
    "content": "Blim"
  },
  ...
]
```

For the current stimulus being queried for a label, the presentation was of the form:

```
{
  "role": "user",
  "content": [
    {
      "type": "text",
      "text": "Based on the preceding examples, what is the label that best describes this?
          Do not give any explanation, and limit your response to exactly one word from
          this list of labels: ['Narp', 'Tovo', 'Feglu', 'Mib', 'Blim', 'Zarn'].\nFeatures:
          [0.73535226, 0.11620602, 0.27974255] -> Label: "
    }
  ]
},
{
  "role": "assistant",
  "content":
},
```

## K SLIDING WINDOW CONVERSATION HISTORY

During the production phase of our IICLL experiments, each response from the model was saved and added to a sliding window of N previous user-model interactions, excluding the initial training data. After preliminary explorations with window sizes of 0, 10, 20, and 50, we determined that a window size of 0 led to degenerate category systems more frequently (see Figures 14 and 15), while the results from a window size of 20 and 50 did not significantly differ from those obtained with a window size of 10. Consequently, we set the window size to 10 for all IICLL experiments. This window was included in the prompt for subsequent stimuli, presented after the initial training data and before each new stimulus, aiming to promote coherence in the model's responses and to crudely mimic the influence of short-term memory that human participants would possess in Xu et al. (2013)'s experiments.

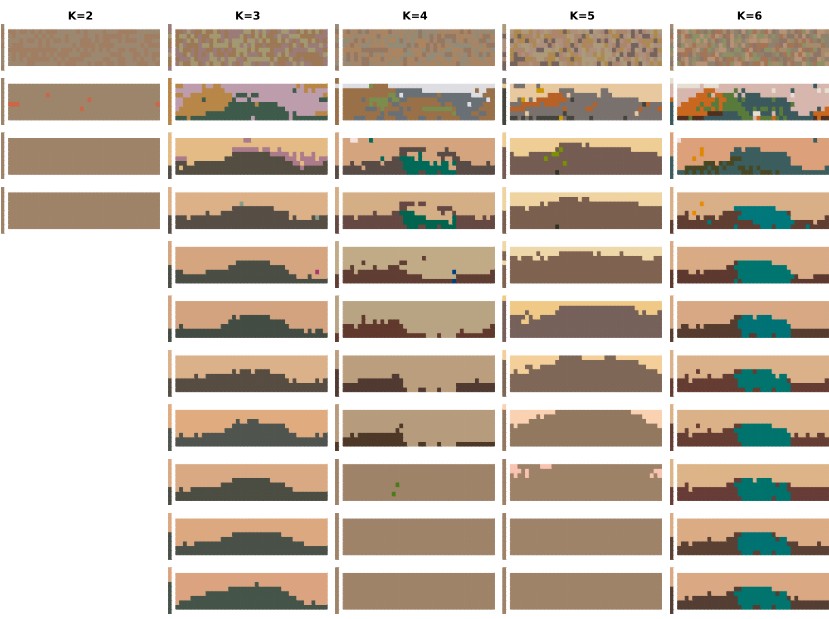

Figure 14: Randomly initialized IICL chains with Llama-3.3-70B-Instruct with a history window of 0.

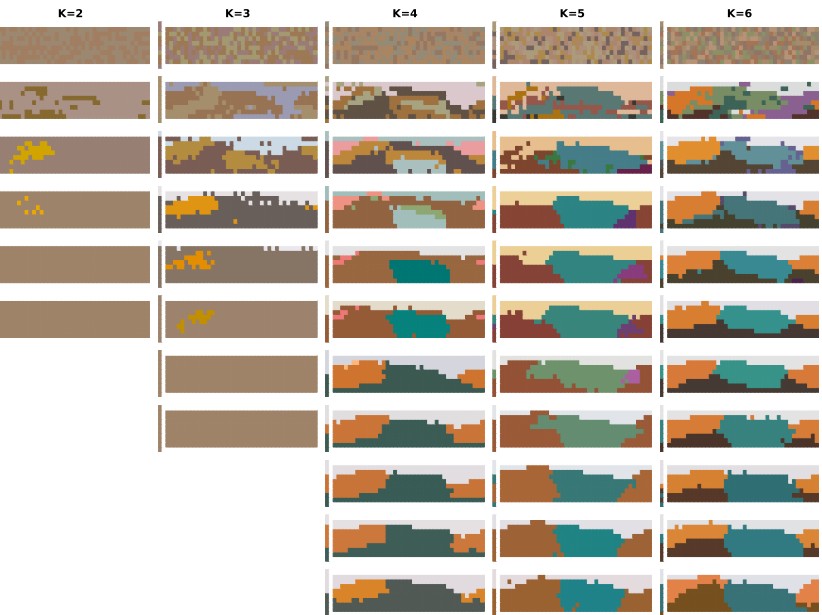

Figure 15: Randomly initialized IICL chains with Gemini-2.0 with a history window of 0.

## L  IICLL WITH SMALLER MODELS

We restricted our main Iterated In-Context Language Learning (IICLL) analysis to models that performed well on the initial English naming task to focus specifically on the evolution towards non-trivial IB-efficiency. This prioritization was based on our expectation that models lacking sufficient inference power, coherence in color naming, or in-context learning capacity would be unable to sustain non-trivial category structure under IL pressure.

In order to confirm this empirically for models with weak initial performance, we ran supplementary IICLL simulations using four models that performed modestly at the English naming task: Gemma 3 4b, Llama 3.2 3B, Qwen 2.5 3B, and OLMo 2 7B (all instruction-tuned). These simulations covered the five vocabulary-size conditions ($k = 2$ through 6) used in the experiments of Xu et al. (2013).

The trajectories of these chains are depicted in Figure 16. Across these 20 chains, only the Gemma model converged to a two-word system (in the $k = 6$ condition), while the remaining 19 chains all converged to degenerate, single-word systems. This confirms that smaller models with limited inference power or in-context learning capacity tend to collapse to degenerate systems when subjected to the noisy transmission pressure of iterated learning.

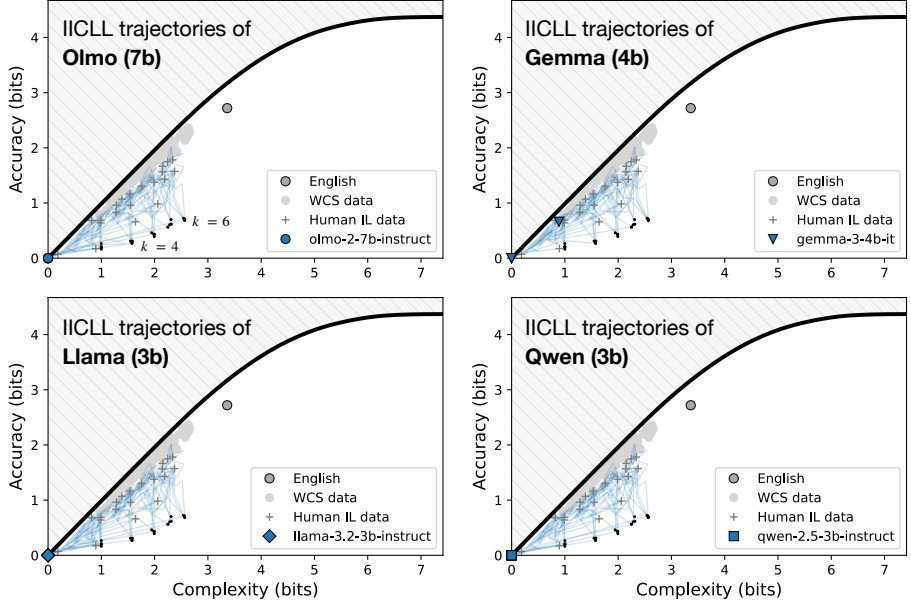

Figure 16: A sample of IICLL trajectories with smaller models from the English naming task. Compare to Figure 3 in the main text.

# M    NEAREST NEIGHBOR BASELINE FOR IICLL

To test whether the semantic category systems that emerge from IICLL are merely simple approximations of feature-based clustering, we conducted a set of baseline simulations using a Nearest Neighbor (NN) classifier in place of an LLM in the IICLL task. In this setup, the NN classifier predicted the category label for each test stimulus based solely on the sRGB features of the provided in-context training examples.

We analyzed performance in the most challenging condition ($k = 14$ allowed terms). The results, presented in Figure 17, show that while the leading open-weight LLMs achieve significantly worse alignment compared to the NN baseline, the leading frontier model, Gemini 2.0, performs significantly better than this baseline on all evaluation metrics (efficiency, IB-alignment, and WCS alignment). In other words, Gemini's evolved categories are significantly more human-like and more IB-like than a simple nearest-neighbor clustering of the input examples.

This result suggests that the ability to acquire a human-like, optimally-compressed semantic representation is (1) not trivial because it is a capability only emergent in the most advanced frontier model, and (2) it cannot be explained by a simpler nearest-neighbor process that only maintains contiguous partitions regardless of their efficiency and IB-alignment.

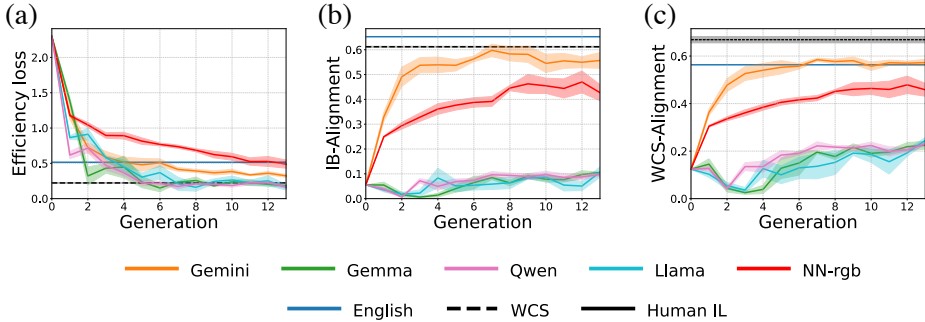

Figure 17: Evolution of LLM systems restricted to the $k = 14$ condition, shown in comparison to the Nearest Neighbor sRGB baseline classifier (NN-rgb). While many models struggle to achieve alignment to IB and human languages comparable to that of the Nearest Neighbor classifier, Gemini achieves greater efficiency (a), alignment with optimal IB systems (b), and alignment with human languages (c) across IICLL generations. Compare to Figure 4.

