# OpenReview forum: "Evolution and compression in LLMs: on the emergence of human-aligned categorization"
_ICLR.cc/2026/Conference — ICLR 2026 Poster_

### Official Review · Reviewer_1kNB · 2025-10-27

**Soundness:** 3
**Presentation:** 3
**Contribution:** 2
**Rating:** 4
**Confidence:** 4

**Summary:**

The present paper investigates whether LLMs are capable of evolving efficient human-aligned semantic systems in the domain of color categorizatzion. The authors replicate two previous human studies on LLMs and find that they vary in their complexity and English-alignment. Larger instruction-tuned models achieved better alignment and IB-efficiency. Finally, they simulate LLMs on cultural evolution of pseudo color-naming systems and find that LLMs iteratively restructure initially random systems towards greater IB-efficiency.

**Strengths:**

Overall, the paper was well written and easy to follow. It builds on prior work with sound methodology, but extends and adapts it for LLMs. The authors consider a large set of models, including models with different properties, different checkpoints, as well as open-source and frontier models.

**Weaknesses:**

The main weakness from my perspective is that novelty and contribution is rather limited. The authors take an existing task and evaluation method and simply run LLMs on it. That might have been enough a few years ago, but it is not  anymore (at least in my opinion). The extension to ICCLLL, which would be the main novelty of the paper, is also fairly straight-forward.

While the authors show some results on how their findings would generalize beyond the color naming domain, more could be done to establish a robust and general pattern.

Even if one would find that LLMs are IB-efficient in their naming systems, would would that mean? What implications would come with that? Or, vice versa, if it were not the case, what would the implications be? From my perspective, these questions deserve some discussion because at present it is a bit unclear to me why I should care.

Minor: p2 rarely -> readily?

**Questions:**

It is interesting that instruction-tuned models achieve better alignment and IB-efficiency. Is that merely an reflection that these models are better at the task? Could one take, for instance, an instruction-tuned and a base model that achieve the same accuracy and look at whether they differ in terms of alignment and IB-efficiency?

To what extend is the observeation that (some) LLMs are IB-efficient a reflection of the fact that they are simply trained on human language?

The authors write "IB-alignment measures the similarity between a system and the nearest optimal IB system. WCS-alignment measures the average alignment between a system and the WCS languages, and English-alignment measures the alignment between a system and English." Are these measures defined somewhere?

---

> ### Author Response · Authors · 2025-11-22
> **Rebuttal response to reviewer's first point**
>
> We thank the reviewer for their comments about our work, and for recognising that our methodology is sound. We believe that the main concerns raised in this review result from some confusion about the nature of our work and contributions. We hope that our response below clarifies these misunderstandings and that the reviewer would be inclined to take this into consideration.
>
>
>
> > “The main weakness from my perspective is that novelty and contribution is rather limited. The authors take an existing task and evaluation method and simply run LLMs on it. That might have been enough a few years ago, but it is not anymore (at least in my opinion). The extension to ICCLLL, which would be the main novelty of the paper, is also fairly straight-forward.”
>
>
> First, we wish to clarify that the IICLL method is not the main novelty or core contribution of our paper. The core novelty and contributions in our work are:
>
> 1. **Conceptual** --- designing a theory-driven study that integrates tools from information theory and cognitive science in order to test a **new research question** about the behavior of LLMs. Specifically, we ask: given that it has previously been shown that human semantic systems evolve under pressure to be IB-efficient, are LLMs capable of acquiring a similar behavior? The motivation for this question is stated in the introduction of the paper (lines 42-45 of the original manuscript): “in order to understand whether LLMs can efficiently communicate with people and adapt to changing environments and communicative needs in a human-like manner, it is crucial to study whether LLMs are capable of structuring meaning according to the same principles that guide humans.”
>
> 2. **Empirical** --- our empirical findings provide important insight for LLM **interpretability** and **alignment**. Specifically, as we state in the abstract, we show that (a) “LLMs vary widely in their complexity and English-alignment, with larger instruction-tuned models achieving better alignment and IB-efficiency”; and (b) “akin to humans, LLMs iteratively restructure initially random systems towards greater IB-efficiency. However, only a model with strongest in-context capabilities (Gemini 2.0) is able to recapitulate the wide range of near-optimal IB-tradeoffs observed in humans, while other state-of-the-art models converge to low-complexity solutions.”
>
> These contributions are not merely a replication of an existing task or a benchmark, as the reviewer seems to argue. Our work provides **new discoveries that advance the field’s knowledge of the implicit learning biases of LLMs**, which should be of broad interest to the ICLR community, and particularly to those who care about AI interpretability and human-AI alignment and communication.
>
> We are therefore quite perplexed by this reviewer’s comment, which focuses only on our methods without addressing our novel conceptual approach to LLMs and our new empirical findings. We hope our response clarifies this issue and that the reviewer would be inclined to evaluate our work based on our actual contributions.

---

> > ### Author Response · Authors · 2025-11-22
> > **Rebuttal response to reviewer's second point**
> >
> > > “While the authors show some results on how their findings would generalize beyond the color naming domain, more could be done to establish a robust and general pattern.”
> >
> > We wish to re-emphasize that our preliminary findings in the Shepard circles domain are intended as an initial sanity check, or proof-of-concept, that the capacity to evolve structured semantic categories is not limited to color. This is not meant to be an in-depth analysis of LLM behavior in this domain, and we are very straightforward about it in the paper. As we explained also in our response to reviewer 5Lw8, a full in-depth analysis of this domain requires a deeper investigation into the underlying representations of the input space. This is an important question of its own for future work, which we are currently looking into, but we think it is beyond the scope of the present paper. While it is always true that “more could be done”,  we believe that our paper already presents a rich set of new findings that would make a valuable contribution to the field. Opening new avenues for future work is another positive outcome in our view.

---

> ### Author Response · Authors · 2025-11-22
> **Rebuttal response to reviewer's third and fourth points**
>
> > “Even if one would find that LLMs are IB-efficient in their naming systems, would would that mean? What implications would come with that? Or, vice versa, if it were not the case, what would the implications be? From my perspective, these questions deserve some discussion because at present it is a bit unclear to me why I should care.”
>
> First, we assume the reviewer meant to ask about the relevance and interest of our work to the **broader ICLR community**, and not to this specific reviewer’s personal view. As we explained earlier, our work provides new empirical findings that **advance the field's knowledge of the implicit learning biases of LLMs and their ability to align with human semantic categories**. These findings should be of broad interest to the ICLR community, and particularly to those who care about **AI interpretability** as well as **human-AI alignment and communication**.
>
> Second, our empirical findings and conclusions are more nuanced than what the reviewer is implying. The first part of our study (English color naming) shows that popular state-of-the-art LLMs actually vary widely in their IB efficiency and complexity, and that the less informationally complex models are also less aligned with English. This means that surprisingly, many LLMs still struggle with this seemingly simple semantic domain, even though they are trained on massive amounts of English data and have billions of parameters. Some models, like Gemma and Gemini, do perform better, and more generally, our results suggest that instruction tuning and model size seem to help. The second part of our study (IICLL) aims to test whether LLMs can acquire an inductive learning bias toward IB-efficiency, beyond imitating their training data. We find that while the best-performing models may have acquired such a bias, they seem to be confined to a low complexity regime, while only a frontier model, Gemini 2.0, can actually capture both the IB-efficiency and wide semantic variation that is observed in humans. This demonstrated that only with exceptionally strong in-context learning capabilities, LLMs may acquire a human-like inductive bias toward IB-efficiency and the ability to evolve on their own human-like semantic systems through cultural transmission.
>
> We recognize that our work pertains more directly to the **science of AI** rather than the engineering of AI. We hope the reviewer can appreciate the value of these scientific findings and insight, and their relevance to the broader ICLR community.
>
>
> > “Minor: p2 rarely -> readily?”
>
> This is not a typo, we mean “rarely”. Although ‘readily’ is also true, we wish to emphasize the unique level of availability and richness of human data available for color naming.

---

> > ### Author Response · Authors · 2025-11-22
> > **Response to reviewer's questions**
> >
> > > “It is interesting that instruction-tuned models achieve better alignment and IB-efficiency. Is that merely an reflection that these models are better at the task? Could one take, for instance, an instruction-tuned and a base model that achieve the same accuracy and look at whether they differ in terms of alignment and IB-efficiency?”
> >
> > We thank the reviewer for the interesting question about comparing instruction-tuned vs. base models of comparable accuracy yielding differing efficiency.  Empirically, we find that there are only two such minimal pairs of models in our sample– Gemma 3 1B vs. its instruction-tuned variant, and likewise for Qwen 2.5 3B– that have comparable accuracy, However, because the values are quite low (below 0.5 bits), and their English alignment scores are also low (as are all base pretrained models), it is unclear what takeaways can be drawn from these data points.
> >
> > We agree with the reviewer that further investigation of this phenomenon would be valuable. Our intuition is that instruction-tuning indeed allows LLMs to transition into a mode that allows them to better simulate participation in a task, and that this may be crucial for performance in our experiments.
> >
> >
> > > “To what extend is the observeation that (some) LLMs are IB-efficient a reflection of the fact that they are simply trained on human language?”
> >
> > The observation that some LLMs exhibit efficiency is not likely a reflection of training on human language, but rather evidence of an acquired functional principle, or in other words, an inductive bias of in-context learning. Indeed, this is one of the major takeaways from our study, as we specifically designed the Iterated in Context Language Learning (IICLL) experiments to investigate this distinction. While LLMs are indeed trained on text data that consists of linguistic forms, the IICLL results demonstrate that they can generalize human-aligned *semantic* representations of color to artificial mappings of stimuli to pseudo-words that were never encountered in their training data. The fact that the models can evolve IB-efficiency in this novel environment indicates they have internalized some bias aligned with the functional pressures that shape human language, rather than just memorizing surface statistics from existing languages.
> >
> >
> > > “The authors write "IB-alignment measures the similarity between a system and the nearest optimal IB system. WCS-alignment measures the average alignment between a system and the WCS languages, and English-alignment measures the alignment between a system and English." Are these measures defined somewhere?”
> >
> > Yes, absolutely, they are defined in the immediately preceding sentences to the text the reviewer quotes (lines 242-246 in the original manuscript, now lines 251-258 in the revision). Specifically, we define misalignment as the Normalized Information Distance (NID) between two systems, and then alignment is defined as $1 - NID$  The precise definition for NID is taken from Kraskov et al. (2005) and Vinh et al. (2010), which we cite for this purpose.

---

> > > ### Comment · Reviewer_1kNB · 2025-11-24
> > >
> > > Thank you for your response. I appreciate the clarifications but my overall assessment remains well reflected by "marginally below the acceptance threshold. But would not mind if paper is accepted".

---

### Official Review · Reviewer_5Lw8 · 2025-10-30

**Soundness:** 4
**Presentation:** 3
**Contribution:** 3
**Rating:** 8
**Confidence:** 4

**Summary:**

The authors investigate large language models’ linguistic capabilities to partition semantically the colour domain and study the efficiency of the resulting systems in terms of the IB framework. Further, they investigate their capabilities to evolve inefficient semantic systems towards more regular categories ((again, in the IB framework sense), both in the colour domain and in the less humanly intuitive Shepard circles one. They show that while many models struggle to recover an efficient colour system, many times exhibiting trivial, low complexity categories, the ones that show efficient, more human-like systems also show the capabilities to evolve a non-efficient artificial system towards more efficient ones via in-context iterated language learning

**Strengths:**

The paper’s biggest contribution is to study an important question in cognitive science, namely the emergence of IB-efficient linguistic systems, in the context of large language models.  The paper is sound, well-written and very easy to follow. The experiments are well designed and multiple models, as well as multiple modalities, are systematically tested. The author’s extension of the in-context iterated learning paradigm of Zhu and Griffiths (2024) is a natural and effective way of studying efficient communication questions in the context of large language models, while the author’s replication of the human experiments from Xu et al. (2013) with large language models offers us an additional perspective on iterated learning as a paradigm to evolve artificial linguistic systems. The authors’ results are both well documented and well presented visually, and are also in line with efficient communication theory.

While previous literature has analysed large language models’ capabilities of recovering human-like semantic systems in the colour domain, or the role of iterated learning in large language models, to my knowledge, this is the first comprehensive study of large language models from an efficient communication point of view.

**Weaknesses:**

While the authors analyse a large number of different families of models and, inside those,  models of many different sizes, they restrict their iterated learning analysis to only the best-performing models at the previous task. It would have been interesting to see if iterated learning might have improved their ability to evolve degenerate systems towards more IB-efficient ones. Similarly, the preliminary work on Shepard Circles is lacking a more quantitative analysis similar to the results in Section 4.2, while a figure showing the evolution of the LLM systems like Figure 3 would have helped quantify the evolution of these systems.

A more general feedback is that the authors do not analyse the possible role for large language models of language use via communication in evolving inefficient linguistic systems toward more human-like ones. While Imel et al. (2025) showed how learnability might be the principal pressure that leads towards efficient systems in the colour and Shepard circles domain, an analysis on the role of both cultural transmission via iterated learning and language use via communication would be a very interesting avenue for future work. More specifically, the framework of Kouwenhoven et al. (2024) might be of interest for the authors.

Smaller points to address would be to fix the citations in the appendices, while some figures (cf. Figure 3) are grainy if zoomed in.

**Questions:**

What do you think might be the role of communication in evolving these systems towards more efficient ones? Do you think that an innate preference for informative, non-trivial systems (which previous efficient communication literature argues is a consequence of language use via communication) is present in the evolution of these artificial languages because of the data these models are trained on?

How do you think discrete domains (e.g. kinship) would influence these results? One could argue that continuous domains like colours and Shepard circles might be simpler to efficiently partition for both humans and large language models than more discrete ones.

---

> ### Author Response · Authors · 2025-11-22
> **Rebuttal response to reviewer's points**
>
> We thank the reviewer for these valuable comments and suggestions, and for the strong positive evaluation of our work. We have addressed the reviewer’s comments in our response below and in our revised submission, and feel that our paper is even stronger now thanks to this process.
>
> > “While the authors analyse a large number of different families of models and, inside those, models of many different sizes, they restrict their iterated learning analysis to only the best-performing models at the previous task. It would have been interesting to see if iterated learning might have improved their ability to evolve degenerate systems towards more IB-efficient ones.”
>
> Our decision to focus in the IL experiments  only on the best-performing models in the initial task was based on the premise that IL tends to amplify existing biases. We expect that models that perform poorly --- specifically those that fail to align with the language on which they were trained --- will quickly collapse to degenerate systems when subjected to the noisy transmission process of iterated learning.  Having said that, we agree with the reviewer that it is important to confirm this intuition. To this end, we ran additional IICLL simulations with four smaller models that performed modestly at the English naming task: instruction-tuned variants of Gemma 3 4B, Llama 3.2 3B, Qwen 2.5 3B and OLMO 27B (see appendix L in the revised version). Across the five conditions ($k=2$ through $6$) of Xu et al. (2013)’s experiments, only Gemma 3 4B converged to a two-word system (in the $k=6$ condition), while the other 19 chains all converged to degenerate, single-word systems. We thank the reviewer for encouraging us to include this important validation.
>
>
> > “Similarly, the preliminary work on Shepard Circles is lacking a more quantitative analysis similar to the results in Section 4.2, while a figure showing the evolution of the LLM systems like Figure 3 would have helped quantify the evolution of these systems.”
>
> We agree with the reviewer that a more comprehensive IB analysis is desirable for the Shepard Circles results. The challenge, however, is that to instantiate an IB model we first need to specify the underlying representation of the domain. In color, for example, the IB model was grounded in a well-established perceptual color space (see Zaslavsky et al., 2018). But for the Shepard circles domain, it is not yet clear what would be the right underlying representational space, and to what extent LLMs and humans may have a similar underlying representation of these stimuli. Determining the correct non-linguistic internal representation for the Shepard stimuli is a challenging open direction for future work (which we are currently working on). We have added a note in the Discussion (lines 498-499) the importance of extending our analysis to more domains.
>
> We wish to reiterate that this section was intended just as a preliminary proof of concept that our IICLL paradigm can be extended beyond color, and that our results have at least the potential to generalize more broadly.
>
>
> > “A more general feedback is that the authors do not analyse the possible role for large language models of language use via communication in evolving inefficient linguistic systems toward more human-like ones. While Imel et al. (2025) showed how learnability might be the principal pressure that leads towards efficient systems in the colour and Shepard circles domain, an analysis on the role of both cultural transmission via iterated learning and language use via communication would be a very interesting avenue for future work. More specifically, the framework of Kouwenhoven et al. (2024) might be of interest for the authors.”
>
> We absolutely agree with the reviewer about this point and are grateful for the reference to Kouwenhoven et al. (2024). Our goal in this paper was to implement the Xu et al. experiment in LLMs and evaluate the result with respect to the empirical human data and the IB framework. We believe this is an important contribution to the field, revealing that an efficiency bias can emerge even without an explicit need for communication. Exploring the role of communication, as the reviewer noted, is an important direction for future work which we are very excited about. We have acknowledged this in the discussion section of our revised submission (lines 490-495) and included a reference to Kouwenhoven et al. (2024).
>
> > “Smaller points to address would be to fix the citations in the appendices, while some figures (cf. Figure 3) are grainy if zoomed in.”
>
> We are indebted to the reviewer for calling our attention to these formatting errors. We apologize for this oversight and have fixed all these issues in the revised submission.

---

> ### Author Response · Authors · 2025-11-22
> **Response to reviewer's questions**
>
> > “What do you think might be the role of communication in evolving these systems towards more efficient ones? Do you think that an innate preference for informative, non-trivial systems (which previous efficient communication literature argues is a consequence of language use via communication) is present in the evolution of these artificial languages because of the data these models are trained on?”
>
> The precise role of the massive training data, and in particular its communicative content, in enabling LLMs to evolve non-trivially efficient systems under Iterated Learning (IL) is an important, but difficult, topic to investigate empirically. Given all the moving parts, the centrality of strong in-context capabilities for running our experiments, and the lack of data transparency for many of the frontier models, it is unclear how we might be able to test this. One thing to note, however, is that modern LLMs are designed to be useful conversational partners, and in a sense, have evolved under this pressure. Therefore, it seems quite plausible that at least some of the frontier models might have an “innate” bias toward informativeness, similar to what Imel et al. (2025) have found in humans. This LLM bias, however, could stem from many aspects, such as their architecture and training methods, in addition to the training data.
>
> We agree with the reviewer that the origins of the bias for efficiency calls out for further investigation. We thank the reviewer for highlighting this important question, and have added it to our Discussion section (lines 495-498).
>
>
> > “How do you think discrete domains (e.g. kinship) would influence these results? One could argue that continuous domains like colours and Shepard circles might be simpler to efficiently partition for both humans and large language models than more discrete ones.”
>
> We completely agree that extending our work to more domains, and especially discrete domains, is another important direction for future work. We are quite optimistic about this prospect given that the IB framework has been successfully applied to domains like personal pronouns (Zaslavsky, Maldonado, and Culbertson, 2021).

---

### Official Review · Reviewer_nH1w · 2025-11-02

**Soundness:** 3
**Presentation:** 3
**Contribution:** 3
**Rating:** 6
**Confidence:** 3

**Summary:**

This paper investigates the categorisation behaviour of LLMs for colour, comparing it with human generated color categories in many languages. It adopts an information bottleneck analysis, testing whether generated colour categories optimally trade off accuracy and complexity. The paper shows that LLMs can differ dramatically in their alignment to English, and the complexity of their categories, with larger models tending to perform better. Second, the paper investigates an iterated learning scheme in which LLMs receive an initially random colour categorisation and over several rounds of in-context classification, converge to a new categorisation. This categorisation can then be assessed for its position on the IB tradeoff plane and compared to human languages. Results show that iterated in context learning progressively improves IB efficiency over iterations, and the strongest models show a similar range of solutions as in humans. Finally, the paper shows that similar ideas can apply beyond the domain of colour, by investigating the categorisation of a class of simple shapes that differ in orientation and size.

**Strengths:**

This paper presents a thorough investigation of colour categorisation behaviour in LLMs. The extent of the experiments, range of models considered, and variety of training hyper parameters investigated (such as training epoch, instruction tuned or not, text vs image prompted, etc) paint a particularly complete picture of the full space of models' abilities in this area.

The results show that, surprisingly, only the most capable models show colour categorisation behaviour matching English. Further, only these models can develop diverse pseudo labels that span the IB tradeoff curve across complexities, similarly to humans.

The basic observations are extended to 'Shepard circles', a particular set of visual stimuli, to show that the proposed ideas are not specific to colour naming.

The paper contains an excellent review of prior work, and motivation for studying colour behaviour as a highly studied instance of human categorisation behaviour.

The experiments appear to be done to a high standard, and the plots are clear and easy to follow.

**Weaknesses:**

There were several points that I struggled to understand which may reflect my own limited knowledge of the area:

-The clusterings seem to more or less subdivide the colour space into a set of contiguous regions with approximately equal size, as one might expect by running k-means on the feature vectors (sRGB values) or any other clustering technique. If this is so, then it may be quite easy for a system to have an implicit bias that would place it near the IB curve--it just needs to approximate k-means or similar. If this is right, then statements like "LLMs can acquire a human-like inductive bias toward optimally-compressed semantic representations, without being trained for this objective" become less remarkable. It is easy to see how in context learning might approximate clustering, and this implicit bias has no clear human-specific content.

-The observation that Gemini 2.0 can recover English alignment from rounds of iterated learning could be consistent with the model inferring that the feature values are sRGB values, and aligning pseudo labels with real colour names. That is, in-context learning may be operating in a ‘task inference’ mode in which it guesses that the features correspond to colours, and then assigns pseudo labels real colour names accordingly. This mechanism would mean that iterated in-context learning is just another measurement of whether an LLM is aligned to English colour categories, not that something fundamental about how its inductive bias aligns with humans.

-The experiments present colours as sRGB triplets, but many other colour spaces are possible. Does the choice of colour space features affect the results?

**Questions:**

How does k-means or another clustering method on the same features compare to the behaviour of the LLMs?

---

> ### Author Response · Authors · 2025-11-22
> **Rebuttal response to first point (1/2)**
>
> We are grateful for the reviewer’s thoughtful comments and questions, and overall strong positive assessment of our work. We believe that our response below, as well as our revised submission, address all the reviewer’s concerns and we feel that our revised paper is stronger now.
>
> > “The clusterings seem to more or less subdivide the colour space into a set of contiguous regions with approximately equal size, as one might expect by running k-means on the feature vectors (sRGB values) or any other clustering technique. If this is so, then it may be quite easy for a system to have an implicit bias that would place it near the IB curve--it just needs to approximate k-means or similar. ”
>
> We would like to start by clarifying that there is an important difference between dividing the space into contiguous regions and finding optimal clustering of the space, as one would get from IB, which is a clustering technique that is in fact quite similar to soft K-means. While these kinds of clustering techniques will indeed yield partitions of the space into contiguous regions, not all contiguous partitions constitute optimal or efficient clustering. Running the IB clustering algorithm, or K-means for this matter, requires an optimization process that repeatedly computes the optimal centroids given a mapping and then computes an optimal mapping based on the centroids, and so on, until convergence.
>
>
> It is unclear to us why or how it may be easy for LLMs to approximate this process in-context, as the reviewer argues. Furthermore, our results suggest that this is not what LLMs are doing in practice. If they were, then by the first generation, models would have to implicitly compute centroids and exhibit contiguous partitions that at least roughly correspond to these centroids. However, the initial generations do not look anything like that (see Fig.13 in Appendix I of the current submission, or Appendix H in the original), and it takes several generations for LLMs (and for humans) to evolve contiguous partitions. After they converge to contiguous partitions, only some models, but crucially not all models, exhibit a significant IB-preference above and beyond alternative contiguous partitions.
>
>
> The rotation analysis we performed (see lines 421-428 and Appendix H in the current version; lines 409-416 and Appendix G in our original submission) targets precisely this issue. This analysis considers as baseline a set of hypothetical systems that were constructed by rotating the emergent color categories along the hue axis, and so all these rotated systems maintain the same degree of contiguity as the original LLM-generated system (see Fig. 12 in the current version). As we discussed in the paper,  in the open-weight models, rotations away from their final emergent system do not always lead to a significant loss in efficiency or WCS alignment. However, rotations away from the systems that Gemini develops does lead to significant losses in efficiency and alignment scores.
>
> This shows that Gemini has a significant preference for IB solutions over alternative contiguous partitions, and because this preference is not expressed in all LLMs, it cannot be a trivial property of these models. More broadly, this findings suggests that only LLMs with powerful enough in-context learning capacities may converge to category systems that partition the semantic space in a way that is specifically aligned to principles of human categorization, and non-trivially optimize lossy compression (in the IB-sense).

---

> > ### Author Response · Authors · 2025-11-22
> > **Rebuttal response to first point (2/2)**
> >
> > > “If this is right, then statements like "LLMs can acquire a human-like inductive bias toward optimally-compressed semantic representations, without being trained for this objective" become less remarkable. It is easy to see how in context learning might approximate clustering, and this implicit bias has no clear human-specific content.”
> >
> > We hope our comment above clarifies why the premise of this statement does not hold. We are also not aware of any existing work that demonstrates that LLMs  can easily approximate K-means optimization through in-context learning. If the reviewer could provide references to published work that shows that, we would be very interested. Otherwise, it seems that this claim might be unsubstantiated.
> >
> > Having said that, we very much appreciate the reviewer’s point about the LLM's behavior possibly being a simple in-context approximation of feature clustering. Perhaps the reviewer had in mind something more akin to a Nearest Neighbor-based classification rather than K-means. We find this possibility a bit more plausible given the literature on LLMs.
> >
> > To investigate this, we conducted a new set of simulations using a Nearest Neighbor (NN) classifier as a baseline for the Iterated In-Context Language Learning (IICLL) task (see Appendix M in the revised version). That is, we replaced the LLM with a NN classifier that predicts the label of each test stimulus based on the label of the nearest sRGB point provided in the in-context training set. We report the results for the most challenging condition of k=14 allowed terms. Crucially, the leading frontier model, Gemini 2.0, achieves significantly better performance in all our metrics for evaluation (efficiency, IB-alignment, and alignment with human systems), while the open source models achieve comparable efficiency but significantly lower alignment scores.
> >
> > This result suggests that the ability to acquire a human-like, optimally-compressed semantic representation is (1) not trivial because it is a capability only emergent in the most advanced frontier model, and (2) it cannot be explained by a simpler nearest-neighbor baseline that only maintains contiguous partitions regardless of IB.

---

> > > ### Author Response · Authors · 2025-11-22
> > > **Rebuttal response to second point**
> > >
> > > > “The observation that Gemini 2.0 can recover English alignment from rounds of iterated learning could be consistent with the model inferring that the feature values are sRGB values, and aligning pseudo labels with real colour names. That is, in-context learning may be operating in a ‘task inference’ mode in which it guesses that the features correspond to colours, and then assigns pseudo labels real colour names accordingly. This mechanism would mean that iterated in-context learning is just another measurement of whether an LLM is aligned to English colour categories, not that something fundamental about how its inductive bias aligns with humans.”
> > >
> > > While interesting, the reviewer’s suggestion is actually inconsistent with our experimental design and results.
> > >
> > > First, we did not argue that “Gemini 2.0 can recover English alignment from rounds of iterated learning,” and as we further explain below, this is actually not implied by our results. As stated in our abstract: “We find that akin to humans, LLMs iteratively restructure initially random systems towards greater IB-efficiency. However, only a model with strongest in-context capabilities (Gemini 2.0) is able to recapitulate the wide range of near-optimal IB-tradeoffs observed in humans.” The latter refers to alignment with the human languages in the World Color Survey, not to English.
> > >
> > > Second, our primary analyses are conducted across a small number of categories, $k=2$ to $k=6$. The English color naming data contains 14 modal  terms and roughly 120 unique terms overall (most of these terms are not modal terms, i.e., they were not used by the majority of participants in response to any given color chips, but they are still allowable terms in English). Given the constraint $k \le 6$, it is not possible for the LLM to infer the 'correct' English color names and align them to pseudo labels. The emergent $k$-term systems must represent a fundamentally different partition of the color space that requires a restructuring that goes beyond simple lexical alignment. This can be seen very clearly in Appendix I, Fig.13, where the converged Gemini systems for $k \le 6$ look nothing like the English color naming system, and more importantly, they contain untranslatable categories. For example, for $k=3$, the model recapitulates a common 3-term system seen in the World Color Survey, which contains a single category that roughly spans colors that English speakers would describe as either blue, green, brown, or black. There is no single word in English that captures this broad color category.
> > >
> > > Third, we specifically added the condition $k=14$, which was not included in the human iterated learning experiment of Xu et al., to test whether LLMs might simply recover the   English system when given a sufficient number of terms. To our surprise, the best performing open-source models (Llama 3.3 70B, Qwen 2.5 32B, and Gemma 3 27B) did not come close to the English system, as can be seen in Fig. 3, and even Gemini reached lower complexity compared to English (and see Fig. 13, which shows that it converges to a system that deviates even from its own version of the English system shown in Fig. 2b).
> > >
> > > Taken together, these points demonstrate that the LLM behavior we observed is not a simple consequence of feature-to-label task inference or recovery of English color names. Thus, we argue that our IICLL design and results provide evidence that LLMs can acquire a human-like inductive learning bias toward IB-optimality, beyond what they may have seen in their training data, if they have exceptionally strong in-context capabilities like Gemini.

---

> > > > ### Author Response · Authors · 2025-11-22
> > > > **Rebuttal response to third point**
> > > >
> > > > > “The experiments present colours as sRGB triplets, but many other colour spaces are possible. Does the choice of colour space features affect the results?”
> > > >
> > > > Yes. We also explored the use of CIELAB coordinates for prompting in the initial English color naming study. In our revised submission (Appendix C), we include mode map visualizations of the resulting systems from this exploration. We observed that performance was generally poor with CIELAB coordinates. This is consistent with the findings of Marjieh et al. (2024), who also found that leading frontier models do not perform well in a similar task when prompted with CIELAB coordinates, while they do perform well when prompted with either sRGB triplets or hexcodes. Based on these observations, , we decided to use sRGB in our IICLL experiments, to focus on what the current models are capable of achieving.
> > > >
> > > > An important direction for future research would be to further study why LLMs fail on CIELAB, while they perform well with sRGB and hexcodes. One potential explanation is that in the models’ training data, colors are much more frequently represented in RGB or hexcodes rather than CIELAB, and therefore RGB or hexcodes seem to provide a better underlying representation of the input colors.

---

> > > > > ### Author Response · Authors · 2025-11-22
> > > > > **Response to reviewer's question**
> > > > >
> > > > > > “Question: How does k-means or another clustering method on the same features compare to the behaviour of the LLMs?”
> > > > >
> > > > >
> > > > > We have addressed this question in our comments above. To summarize:
> > > > >
> > > > > 1. We considered two baselines that maintain contiguous categories: a rotation analysis, included in our initial submission, and a new nearest-neighbor clustering baseline, which we implemented in response to the reviewer’s comments. Gemini significantly outperforms both baselines (see appendices H and M in the revised version), showing that our findings cannot be trivially explained by any clustering method that simply maintains contiguous categories without any additional pressure for IB optimality.
> > > > >
> > > > > 2. K-means would yield systems that are highly correlated with the IB systems, given the similarity and theoretical links between these two clustering methods (IB is somewhat similar to soft K-means). However, as we noted, there is no evidence that LLMs can approximate K-means optimization via in-context learning, and our results suggest that this is not what the models are doing in-context. Over generations, some models, but crucially not all of them, gradually restructure their systems toward significantly IB-efficient systems, and only Gemini can also capture the wide range of complexities seen across human languages. Therefore, this emergent property of the model cannot be trivial as most LLMs struggle to achieve it.
> > > > >
> > > > > We would like to thank the reviewer again for the valuable comments that helped us clarify and strengthen our paper. We hope the reviewer is satisfied with our response, but if there are any persisting concerns, we would appreciate it if the reviewer could let us know as we’d be happy to further discuss.

---

### Meta-Review · Area_Chair_seKv · 2026-01-01

**Summary:**

This paper investigates the categorisation behaviour of LLMs for colour, comparing it with human-generated colour categories across many languages. It adopts an information bottleneck analysis to test whether the generated colour categories optimally trade off accuracy and complexity. The paper shows that LLMs can differ dramatically in their alignment to English and the complexity of their categories, with larger models tending to perform better. Second, the paper investigates an iterated learning scheme in which LLMs receive an initially random colour categorisation and, over several rounds of in-context classification, converge to a new categorisation. This categorisation can then be assessed for its position on the IB tradeoff plane and compared to human languages. Results show that iterated in-context learning progressively improves IB efficiency over iterations, and the strongest models produce a similar range of solutions to those produced by humans. Finally, the paper shows that similar ideas can apply beyond the domain of colour by investigating the categorisation of a class of simple shapes that differ in orientation and size.

The paper is well-written, with comprehensible visualisations. The concerns raised by the reviewers can be summarised as follows:

t simply be approximating k-means or basic clustering on RGB features, making contiguous partitions without genuine IB optimisation. This would make the findings less remarkable since in-context learning could be easily approximated by clustering. The authors conducted two new analyses which in my opinion     Authors conducted two new analyses which show that this is not the case. Rotation analysis showed that rotating emergent color categories maintains contiguity but only Gemini's system shows significant efficiency/alignment loss when rotated, proving preference for IB solutions over mere contiguous partitions. The authors also added a  nearest neighbor baseline  which showed that Gemini 2.0 significantly outperforms NN classifier on all metrics, while open-source models achieve comparable efficiency but lower alignment scores.

2. Gemini might simply infer that RGB values represent colors and align pseudo-labels with real English color names, rather than demonstrating fundamental inductive bias alignment. The authors showed this explanation is inconsistent with their design.

3. The work merely applies existing tasks to LLMs without sufficient novelty; the IICLL extension is straightforward. The authors clarified their contribution is conceptual, theory-driven integration of information theory and cognitive science to test whether LLMs acquire human-like IB efficiency bias and empirical providing new findings about LLM variation in complexity/alignment, and that only frontier models (Gemini 2.0) can recapitulate human-like IB-tradeoffs

In my opinion, the authors successfully addressed technical concerns through new analyses while defending their work's conceptual contribution, demonstrating that only the most advanced LLMs acquire human-like inductive biases toward IB-optimal semantic systems, which has implications for AI interpretability and alignment.

**Reviewer Concerns:**

There are no outstanding concerns in my opinion.

**Reviewer Scores:**

Reviewer 1kNB left a comment saying they didn't want to change their score but were OK if the paper were accepted. Reviewer 5Lw8 was positive (initial score was 8), and reviewer nH1w gave a 6, which could have moved upwards, as the authors added experiments to address their concerns.

---

### Decision · Program_Chairs · 2026-01-26

Accept (Poster)